# POLO: Preference-Guided Multi-Turn Reinforcement Learning for Lead Optimization

## Abstract

Lead optimization in drug discovery requires efficiently navigating vast chemical space through iterative cycles to enhance molecular properties while preserving structural similarity to the original lead compound. Despite recent advances, traditional optimization methods struggle with sample efficiency—achieving good optimization performance with limited oracle evaluations. Large Language Models (LLMs) provide a promising approach through their in-context learning and instruction following capabilities, which align naturally with these iterative processes. However, existing LLM-based methods fail to leverage this strength, treating each optimization step independently. To address this, we present POLO (**P**reference-guided multi-turn **O**ptimization for **L**ead **O**ptimization), which enables LLMs to learn from complete optimization trajectories rather than isolated steps. At its core, POLO introduces Preference-Guided Policy Optimization (PGPO), a novel reinforcement learning algorithm that extracts learning signals at two complementary levels: trajectory-level optimization reinforces successful strategies, while turn-level preference learning provides dense comparative feedback by ranking intermediate molecules within each trajectory. Through this dual-level learning from every intermediate evaluation, POLO achieves superior sample efficiency by fully exploiting each costly oracle call. Extensive experiments demonstrate that POLO achieves **84%** average success rate on single-property tasks (**2.3×** better than baselines) and **50%** on multi-property tasks using only 500 oracle evaluations, significantly advancing the state-of-the-art in sample-efficient molecular optimization. Our code and data are available at `https://anonymous.4open.science/r/POLO-6861/`.

## 1 Introduction

Lead optimization is one of the most critical yet challenging stages in drug discovery, where promising molecular candidates undergo iterative cycles of Design-Make-Test-Analyze (DMTA) to improve multiple key properties while preserving structural similarity to the original lead compound (Plowright et al., 2012; Wesolowski & Brown, 2016). This sequential process, where each modification builds on insights from previous experiments, faces challenges from the vastness of chemical space and the high costs of experimental evaluations (Bleicher et al., 2003; Rácz et al., 2025). Despite advances in computational methods including Genetic Algorithms (GA) (Jensen, 2019), Bayesian optimization (Korovina et al., 2020), and Reinforcement Learning (RL) (Olivecrona et al., 2017), existing approaches often remain sample-inefficient, struggling to achieve strong performance under limited oracle budgets (Gao et al., 2022; Guo & Schwaller, 2024).

Large Language Models (LLMs) offer a transformative opportunity for molecular optimization. Beyond generating valid molecules, LLMs possess unique advantages crucial for sample-efficient lead optimization. Their in-context learning capabilities enable them to leverage optimization histories, while their instruction-following abilities, developed through pre-training on vast chemical corpora, allow them to execute complex molecular transformations. Recent work starts to explore these capabilities through direct prompting (Guo et al., 2023; Liu et al., 2024a), instruction fine-tuning (Ye et al., 2025; Dey et al., 2025), and integration within evolutionary frameworks (Wang et al., 2024a). However, these approaches treat each optimization step independently, processing molecule modifications and evaluations in isolation and discarding results after use. This overlooks the iterative nature of lead optimization, where both successful and failed attempts provide critical information for guiding subsequent modifications (Guo & Schwaller, 2024). As illustrated in Figure 1, existing approaches generally fall into two paradigms: (1) single-turn RL that optimize molecules

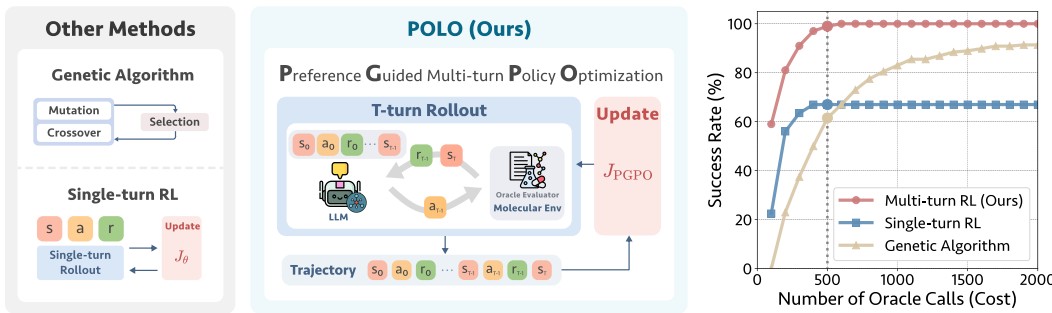

Figure 1: **Comparison of molecular optimization paradigms. Left:** Traditional approaches include genetic algorithms that rely on random mutation and crossover operators, and single-turn RL that optimizes based on isolated tuples with sparse feedback. **Center:** POLO employs Preference-Guided Multi-turn Policy Optimization, where an LLM agent iteratively refines molecules through T-turn rollouts. **Right:** Performance on `plogP` task demonstrates POLO's superior sample efficiency. Our PGPO approach (red) rapidly achieves a near-perfect success rate, while single-turn RL (blue) plateaus early and genetic algorithms (brown) converge slowly, highlighting the advantage of learning from complete optimization trajectories.

independently without leveraging historical information, and (2) Genetic Algorithms that maintain populations of molecules but employ LLMs as mutation operators. Neither paradigm fully exploits LLMs' potential to accumulate and reason over past experience.

Given these limitations, efficient lead optimization requires enabling LLMs to learn from complete optimization trajectories. Multi-turn reinforcement learning naturally aligns with the iterative nature of lead optimization, allowing RL agents to build upon previous attempts and develop long-term strategies (Wang et al., 2024b; Shani et al., 2024). However, standard multi-turn RL suffers from sparse rewards, learning only from final outcomes while discarding valuable data from intermediate steps. In budget-constrained settings, this represents a significant inefficiency as expensive intermediate evaluations are not fully exploited for learning. Our key insight is that optimization trajectories contain rich learning signals at every intermediate step that can significantly improve sample efficiency, turning every costly oracle call into a learning opportunity.

In this work, we present POLO (**P**reference-guided multi-turn **O**ptimization for **L**ead **O**ptimization), a framework that transforms general-purpose LLMs into sample-efficient molecular optimization specialists. At its core, POLO introduces **Preference-Guided Policy Optimization (PGPO)**, a novel reinforcement learning algorithm that realizes our key insight: extracting learning signals at two complementary levels. Trajectory-level optimization reinforces successful optimization strategies, while turn-level preference learning ranks intermediate molecules to provide dense comparative feedback about which modifications improve molecular properties. By learning from every molecular evaluation throughout the trajectory, this dual-level approach maximizes the value extracted from each costly oracle call. Extensive experiments across diverse molecular optimization tasks demonstrate that POLO achieves **84%** average success rate on single-property tasks (**2.3×** better than the best baseline) and **50%** on challenging multi-property tasks using only 500 oracle evaluations, establishing a new state-of-the-art in sample-efficient molecular optimization.

## 2 METHOD

We introduce POLO, a multi-turn RL framework that transforms general-purpose LLMs into sample-efficient molecular optimization specialists through Preference-Guided Policy Optimization (PGPO). The following sections will first define the lead optimization problem (Section 2.1), motivate our proposed multi-turn approach (Section 2.2), formalize our method as a Markov Decision Process (MDP) (Section 2.3), and detail the PGPO algorithm (Section 2.4). We also develop critical supporting components: similarity-aware instruction tuning that provides the chemical foundation (Appendix E) and an evolutionary inference strategy that amplifies performance (Appendix D).

### 2.1 PROBLEM DEFINITION

We represent each molecule as an element $m$ within a vast chemical space $\mathcal{M}$. These molecules are evaluated by a set of black-box property oracles $F_i : \mathcal{M} \to \mathbb{R}$ (e.g., binding affinity, synthetic accessibility) and compared using a Tanimoto similarity function $\text{sim}(\cdot, \cdot)$.

In this work, we focus on the specific problem of lead optimization, where a given lead compound is modified to enhance its key properties under a limited budget. Let the initial lead molecule be $m \in \mathcal{M}$. We denote the set of $n$ property oracles as $\{F_i\}$, their corresponding weights as $\{w_i\}$, the similarity threshold as $\gamma$, and the total oracle budget as $B$. Therefore, our objective is to find an optimized molecule $m'$ that solves the following constrained maximization problem:

$$\max_{m' \in \mathcal{M}} \sum_{i=1}^{n} w_i F_i(m') \quad \text{s.t.} \quad \text{sim}(m, m') \geq \gamma, \quad \sum_{i=1}^{n} \text{oracle\_calls}(F_i) \leq B. \tag{1}$$

## 2.2 MOTIVATION

Lead optimization is a costly and resource-intensive process, largely because current methods fail to explore chemical space efficiently. We argue that LLMs, with their in-context learning capabilities, offer a transformative alternative by treating optimization as a multi-turn conversation. In this paradigm, the model builds on knowledge gained from each evaluation to guide subsequent molecular modifications, much like a human chemist learns from previous synthesis attempt. Yet most existing computational approaches fail to capture this iterative nature, instead treating each molecule in isolation as shown in Figure 1, and thereby missing a key ingredient of successful lead optimization.

**Genetic approaches lack learning capability.** Methods like Graph-GA (Jensen, 2019) and MOLLEO (Wang et al., 2024a) employ genetic operators to generate molecular variations. While maintaining populations across generations, these approaches treat each modification as a random mutation without learning from the optimization landscape. As shown in Figure 1 (right), Graph-GA requires nearly 2000 oracle calls to slowly climb toward 90% success rate. This slow convergence stems from blind trial-and-error: the algorithm wastes hundreds of evaluations exploring unpromising directions without learning from past attempts.

**Single-turn methods discard valuable feedback.** Similarly, single-turn approaches like Reinvent (Olivecrona et al., 2017) and most LLM-based methods (Dey et al., 2025; Ye et al., 2025) optimize each molecule independently. Despite receiving expensive oracle feedback, they cannot leverage information from previous attempts: each generation starts fresh, ignorant of what worked or failed before. Figure 1 demonstrates this inefficiency with single-turn RL achieving only 67% success rate, plateauing early without the ability to develop iterative optimization strategies.

**Multi-turn optimization as the solution.** The performance gap between these approaches and our multi-turn method (Figure 1, red curve) reveals the power of trajectory-based learning. By maintaining complete optimization histories and learning from every evaluation, POLO rapidly achieves near-perfect performance with minimal oracle calls. Each modification builds on accumulated knowledge, transforming random exploration into strategic optimization. This mirrors how human chemists iterate, learning from each synthesis to inform the next design.

## 2.3 LEAD OPTIMIZATION AS A MULTI-TURN MDP

To support trajectory-based learning, we model lead optimization as a multi-turn Markov Decision Process (MDP). In this formulation, an LLM agent iteratively generates candidate molecules, using feedback from previous modifications to guide its decisions. By keeping a history of all prior attempts, the agent can identify successful strategies and adjust accordingly. This process shifts the LLM from acting as a one-shot generator to a strategic decision-maker that improves through experience.

Formally, we define the MDP as $\mathcal{M} = \langle \mathcal{S}, \mathcal{A}, P, R \rangle$:

- **State Space** $\mathcal{S}$: At each turn $t$, the state $s_t$ encodes the complete conversational context available to the agent. This includes three key components: (1) task instructions and optimization objectives (as detailed in Appendix E), (2) all proposed molecules $(m_0, \ldots, m_t)$ where $m_0$ is the initial lead, and (3) oracle evaluations $(r_0, \ldots, r_{t-1})$ for previously generated molecules.

- **Action Space** $\mathcal{A}$: An action $a_t \sim \pi_\theta(\cdot|s_t)$ is the agent's response, formatted as a structured output: `<think>...</think><answer>...</answer>`. The `<think>` block contains the agent's reasoning process, while the `<answer>` block contains the new candidate SMILES string.

- **Transition and Reward** $P, R$: Upon receiving action $a_t$, the environment executes three steps: (1) extracts the SMILES string $m_{t+1}$ from the `<answer>` tag, (2) evaluates it using black-box oracles to compute reward $r_t$ (detailed in Appendix B), and (3) constructs the next state $s_{t+1}$ by appending $m_{t+1}$ and $r_t$ to the conversational history. This transition is denoted as $(r_t, s_{t+1}) \sim P(\cdot|s_t, a_t)$.

Figure 2: **Overview of Preference-Guided Policy Optimization (PGPO).** The LLM agent performs multi-turn molecular optimization, iteratively refining molecules based on oracle feedback while maintaining structural similarity. PGPO employs dual-level learning: (1) **trajectory-level optimization** uses cumulative rewards $R$ from the entire optimization path to reinforce successful strategies, and (2) **turn-level preference learning** performs listwise ranking of all molecules within each trajectory, creating multiple pairwise training signals to maximize learning from limited oracle calls.

An optimization episode unfolds as trajectory $\tau = (s_0, a_0, r_0, \ldots, a_{T-1}, r_{T-1}, s_T)$, terminating when the generated molecule achieves the target property thresholds or reaches horizon $T$ (typically 5 turns). Each state $s_t$ contains the complete history up to turn $t$, providing increasingly rich context that guides subsequent molecular generation. This multi-turn structure enables us to extract learning signals from every intermediate evaluation, not just the final outcome, forming the foundation of our preference-guided optimization approach described next.

## 2.4 PREFERENCE-GUIDED POLICY OPTIMIZATION

The multi-turn MDP provides a framework for strategic planning, but learning within this setting faces a key challenge. Oracle feedback is both sparse and expensive. Often the agent receives only a single cumulative reward at the end of a long trajectory, making it difficult to identify which specific modifications were beneficial. To overcome this limitation and take full advantage of every oracle evaluation, we introduce Preference-Guided Policy Optimization (PGPO), which combines two complementary learning signals at different levels:

$$J_{\text{PGPO}}(\theta) = \underbrace{J_{\text{traj}}(\theta)}_{\text{trajectory-level}} + \lambda_{\text{pref}} \underbrace{J_{\text{pref}}(\theta)}_{\text{turn-level}}, \tag{2}$$

where $\lambda_{\text{pref}}$ balances the two forms of guidance as follows:

- **Trajectory-level:** This component acts as a strategic guide by optimizing cumulative reward $R(\tau)$. Rather than focusing on individual molecules, it reinforces entire optimization paths that successfully reach the target. This teaches the agent what constitutes an effective multi-turn strategy, preventing it from pursuing modifications that seem promising initially but lead to dead ends.

- **Turn-level:** In contrast, this component provides fine-grained feedback by learning from the relative performance of molecules generated within each trajectory. When a trajectory produces molecules with varying oracle scores, we can extract preferences (e.g., molecule A scores higher than B), teaching the agent which chemical modifications are more promising.

Together, this dual-level approach transforms the challenging problem of navigating high-dimensional chemical space into a more tractable learning task. The following sections detail each component's formulation and implementation.

### 2.4.1 TRAJECTORY-LEVEL POLICY OPTIMIZATION

The trajectory-level component optimizes entire molecular optimization paths rather than individual modifications. Following recent advances in multi-turn agent learning (Wang et al., 2025), we treat each trajectory $\tau = (s_0, a_0, r_0, \ldots, s_T)$ as the fundamental learning unit and maximize expected

cumulative reward using Proximal Policy Optimization (PPO) (Schulman et al., 2017):

$$J_{\text{traj}}(\theta) = \mathbb{E}_{\tau \sim \pi_\theta} \left[ \sum_{t=0}^{T-1} \min \left\{ \rho_t(\theta) \hat{A}_t, \text{clip}(\rho_t(\theta), 1 - \epsilon, 1 + \epsilon) \hat{A}_t \right\} \right], \tag{3}$$

where $\rho_t(\theta) = \frac{\pi_\theta(a_t|s_t)}{\pi_{\theta_{\text{old}}}(a_t|s_t)}$ is the importance sampling ratio between the current policy $\pi_\theta$ and the old policy $\pi_{\theta_{\text{old}}}$ (from which the trajectory was collected). The advantage estimates $\hat{A}_t$ quantify how much better action $a_t$ is compared to the average action at state $s_t$, estimated from the trajectory rewards $R(\tau) = \sum_{t=0}^{T-1} r_t$. Specifically, each turn-level reward $r_t$ balances property optimization with similarity constraints:

$$r_t = \sum_{i=1}^{n} w_i \Delta F_i(m_{t+1}, m_t) - \max\{0, \gamma - \text{sim}(m_0, m_{t+1})\}, \tag{4}$$

where $m_{t+1}$ is extracted from action $a_t$, $\Delta F_i(m_{t+1}, m_t) = F_i(m_{t+1}) - F_i(m_t)$ measures the change in property $i$, and the second term penalizes similarity violations below threshold $\gamma$. In practice, we augment this with asymmetric scaling and validity checks to encourage exploration (see Appendix B).

### 2.4.2 Turn-Level Preference Learning

The key innovation of PGPO is augmenting trajectory-level optimization with dense, turn-level preference signals. Within each trajectory, different turns produce molecules with varying oracle scores, creating an implicit ranking that reveals which modifications are more promising. Rather than discarding this rich preference information, we extract it to provide fine-grained learning signals.

Following Direct Preference Optimization (DPO) (Rafailov et al., 2023), we can learn from these preferences without explicit reward modeling. DPO parameterizes preferences through the log-ratio between the current policy and a reference policy:

$$\psi_t = \beta \log \frac{\pi_\theta(a_t|s_t)}{\pi_{\text{ref}}(a_t|s_t)},$$

where $\psi_t$ measures how much more likely the current policy is to take action $a_t$ compared to the reference. To ensure our agent starts with the ability to make chemically valid modifications, we initialize $\pi_{\text{ref}}$ by fine-tuning on MolOptIns, a dataset of 500K high-quality molecular modifications with high structural similarity (see Appendix E for details). This reference policy serves dual purposes: it provides a strong chemical prior for valid modifications and acts as a regularizer to prevent $\pi_\theta$ from deviating too far from chemically sensible solutions.

Unlike standard DPO which handles single preference pairs, our multi-turn trajectories naturally contain multiple molecules, enabling richer preference learning. With up to $\binom{T}{2}$ potential pairs per trajectory, we adopt a listwise ranking perspective that considers all molecules simultaneously.

Our preference objective aggregates comparisons across selected pairs $(i, j)$ where $r_j > r_i$:

$$J_{\text{pref}}(\theta) = -\mathbb{E}_{\tau \sim \mathcal{D}} \left[ \sum_{(i,j) \in \mathcal{P}} \Lambda_{i,j} \cdot \log \left( 1 + \exp \left( -(\psi_j - \psi_i) \right) \right) \right], \tag{5}$$

where $\mathcal{P}$ denotes selected pairs where molecule $j$ outperforms molecule $i$ ($r_j > r_i$). When $\psi_j > \psi_i$ (the model correctly ranks the better modification higher), the loss approaches zero, encouraging the policy to prefer better molecules.

While this objective captures all pairwise preferences, not all pairs are equally informative—correcting the ranking between very similar molecules is less important than fixing large ranking errors. Drawing from the Learning-to-Rank (LTR) literature (Burges et al., 2006; Wang et al., 2018), we introduce Lambda weights to prioritize the most impactful comparisons:

$$\Lambda_{i,j} = |G(r_i) - G(r_j)| \cdot \left| \frac{1}{D(\rho_i)} - \frac{1}{D(\rho_j)} \right|, \tag{6}$$

where $G(r) = 2^r - 1$ is the gain function, $D(\rho) = \log(1 + \rho)$ is the rank discount, and $\rho_i$ denotes the rank position determined by sorting all molecules within the trajectory by their environment rewards

Table 1: **Overall Performance of Single Property Optimization.** For each task, the best baseline performance is underlined and the best overall performance is in **bold**.

| Model | QED | | | plogP | | | DRD2 | | | JNK3 | | | SA | | |
|---|---|---|---|---|---|---|---|---|---|---|---|---|---|---|---|
| | SR (%) | Sim | RI | SR (%) | Sim | RI | SR (%) | Sim | RI | SR (%) | Sim | RI | SR (%) | Sim | RI |
| **Without LLMs** | | | | | | | | | | | | | | | |
| Graph-GA | 59.50 | 0.49 | 0.13 | 61.50 | 0.49 | 9.64 | 34.00 | 0.56 | 5.13 | 2.00 | 0.59 | 1.73 | 46.00 | 0.48 | 0.20 |
| QMO | 16.50 | 0.53 | 0.16 | 8.00 | 0.55 | 5.87 | 9.50 | 0.51 | 3.91 | 0.50 | 0.56 | 1.20 | 6.50 | 0.52 | 0.15 |
| Reinvent 4 | 33.50 | 0.50 | 0.14 | 51.50 | 0.48 | 10.37 | 34.50 | 0.50 | 8.71 | 1.00 | 0.51 | 1.91 | 18.50 | 0.47 | 0.21 |
| **General-purpose LLMs** | | | | | | | | | | | | | | | |
| Qwen2.5-7B | 59.00 | 0.56 | 0.20 | 23.50 | 0.63 | 9.35 | 13.50 | 0.64 | 4.92 | 0.00 | 0.63 | 1.46 | 6.00 | 0.54 | 0.22 |
| Llama3.1-8B | 46.50 | 0.67 | 0.17 | 29.00 | 0.64 | 13.20 | 25.50 | 0.68 | 8.14 | 1.50 | 0.67 | 2.62 | 15.00 | 0.66 | 0.19 |
| **Task-Specific LLMs** | | | | | | | | | | | | | | | |
| MOLLEO | 15.00 | 0.68 | 0.08 | 12.50 | 0.70 | 3.37 | 4.00 | 0.78 | 0.63 | 0.00 | 0.88 | 0.04 | 8.00 | 0.81 | 0.02 |
| LlaSMol | 33.50 | 0.46 | 0.14 | 23.50 | 0.49 | 8.99 | 14.00 | 0.47 | 5.26 | 0.00 | 0.44 | 0.71 | 19.00 | 0.51 | 0.19 |
| ChemLLM | 29.00 | 0.51 | 0.12 | 19.50 | 0.54 | 6.15 | 15.00 | 0.56 | 6.20 | 0.00 | 0.51 | 0.98 | 12.00 | 0.57 | 0.13 |
| PEIT-LLM | 52.50 | 0.45 | 0.15 | 51.00 | 0.48 | 12.09 | 21.50 | 0.47 | 6.02 | 0.00 | 0.49 | 0.88 | 19.50 | 0.45 | 0.19 |
| GeLLM³O | 61.50 | 0.56 | 0.19 | 57.00 | 0.52 | 12.80 | 49.00 | 0.56 | 11.16 | 0.00 | 0.55 | 1.19 | 14.50 | 0.57 | 0.16 |
| POLO (**Ours**) | **91.00** | 0.49 | 0.23 | **99.00** | 0.49 | 25.91 | **97.00** | 0.49 | 16.69 | **81.00** | 0.47 | 10.06 | **53.50** | 0.51 | 0.30 |

$r_i$. This weighting scheme prioritizes pairs with large reward differences (first term) that are currently misranked (second term).

The power of turn-level preference learning lies in its sample efficiency. While standard RL extracts only $\mathcal{O}(N)$ learning signals from $N$ trajectories (one cumulative reward per trajectory), our approach generates up to $\mathcal{O}(NT^2)$ pairwise comparisons from the same data. This quadratic increase enables us to extract maximal value from every expensive oracle call, directly addressing the fundamental bottleneck of limited feedback in molecular optimization. Importantly, these comparisons require no additional oracle evaluations, sharing the same trajectory rollouts.

To further enhance performance, we employ an evolutionary inference strategy that combines our trained policy with population-based search. Rather than running independent rollouts, we maintain an Elite Pool of high-performing molecules accumulated over successive generations. Our PGPO agent then generates targeted modifications based on its trained policy. By combining policy-driven exploitation with population-based exploration, the framework yields consistent performance gains and demonstrates particular robustness in challenging multi-objective optimization settings. We provide complete implementation details in Appendix D.

## 3 EXPERIMENTS

### 3.1 EXPERIMENTAL SETUP

**Baselines.** We evaluate POLO against 10 baseline methods across three categories: **(1) Traditional methods:** Graph-GA (Jensen, 2019), QMO (Hoffman et al., 2022), and Reinvent 4 (Loeffler et al., 2024), representing state-of-the-art non-LLM approaches; **(2) General-purpose LLMs:** Qwen2.5-7B (Yang et al., 2025) and Llama3.1-8B (Dubey et al., 2024), which we prompt with molecular optimization instructions; **(3) Task-specific LLMs:** MOLLEO (Wang et al., 2024a), LlaSMol (Yu et al., 2024), ChemLLM (Zhang et al., 2024), PEIT-LLM (Lin et al., 2024), and GeLLM³O (Dey et al., 2025), which are specifically trained or fine-tuned for molecular tasks. Notably, POLO uses Qwen2.5-1.5B-Instruct, a 1.5B parameter model, while most LLM baselines use 7-8B parameter models. Detailed baseline descriptions and implementation protocols are provided in Appendix F.

**Tasks and Constraints.** We evaluate on 5 single-property (QED, plogP, DRD2, JNK3, SA) and 5 multi-property optimization tasks using 200 lead molecules randomly sampled from ZINC-250k (Irwin & Shoichet, 2005). Following real-world drug discovery constraints, we enforce: (1) Tanimoto similarity $\gamma \geq 0.4$ to preserve lead structures, and (2) oracle budget $B = 500$ calls per optimization.

**Evaluation Metrics.** Following prior work (Dey et al., 2025), we employ three complementary metrics: **(1) Success Rate (SR):** percentage of molecules achieving target improvements while maintaining similarity; **(2) Similarity (Sim):** average Tanimoto similarity between lead and optimized molecules; **(3) Relative Improvement (RI):** average relative improvement across target properties. Detailed formulations and task-specific success criteria are provided in Appendix H.

Table 2: **Overall Performance of Multiple Property Optimization.** For each task, the best baseline performance is underlined and the best overall performance is in **bold**.

| Model | QED + plogP | | | plogP + DRD2 | | | QED + SA | | | DRD2 + SA | | | DRD2 + QED + plogP | | |
|---|---|---|---|---|---|---|---|---|---|---|---|---|---|---|---|
| | SR (%) | Sim | RI | SR (%) | Sim | RI | SR (%) | Sim | RI | SR (%) | Sim | RI | SR (%) | Sim | RI |
| **Without LLMs** | | | | | | | | | | | | | | | |
| Graph-GA | 8.00 | 0.50 | 4.11 | 2.50 | 0.48 | 6.61 | 8.00 | 0.52 | 0.12 | 0.00 | 0.53 | 4.15 | 0.00 | 0.51 | 3.79 |
| QMO | 3.00 | 0.52 | 6.40 | 2.00 | 0.51 | 4.20 | 1.50 | 0.53 | 0.11 | 1.00 | 0.50 | 2.49 | 0.00 | 0.52 | 1.14 |
| Reinvent 4 | 6.00 | 0.49 | 8.60 | 50.00 | 0.48 | 7.94 | 17.00 | 0.66 | 0.05 | 37.50 | 0.50 | 5.54 | 3.00 | 0.51 | 4.28 |
| **General-purpose LLMs** | | | | | | | | | | | | | | | |
| Qwen2.5-7B | 5.00 | 0.60 | 7.05 | 4.00 | 0.60 | 5.47 | 8.50 | 0.62 | 0.08 | 1.00 | 0.59 | 3.44 | 0.50 | 0.58 | 4.29 |
| Llama3.1-8B | 7.50 | 0.53 | 6.18 | 5.00 | 0.55 | 5.21 | 13.50 | 0.55 | 0.14 | 7.50 | 0.57 | 4.34 | 1.00 | 0.55 | 3.44 |
| **Task-Specific LLMs** | | | | | | | | | | | | | | | |
| LlaSMol | 8.50 | 0.45 | 2.85 | 4.00 | 0.45 | 3.17 | 7.50 | 0.43 | 0.11 | 6.50 | 0.47 | 2.14 | 0.00 | 0.45 | 1.79 |
| ChemLLM | 4.00 | 0.53 | 5.66 | 11.00 | 0.53 | 5.48 | 11.00 | 0.54 | 0.12 | 6.00 | 0.53 | 4.14 | 0.00 | 0.52 | 2.85 |
| PEIT-LLM | 9.50 | 0.42 | 5.51 | 16.50 | 0.45 | 4.94 | 25.00 | 0.43 | 0.17 | 16.00 | 0.46 | 3.94 | 1.50 | 0.42 | 3.02 |
| GeLLM$^3$O | 9.50 | 0.56 | 5.22 | 16.50 | 0.57 | 5.34 | 21.00 | 0.55 | 0.15 | 15.50 | 0.57 | 4.34 | 0.00 | 0.57 | 3.46 |
| **POLO (Ours)** | **55.00** | 0.51 | 10.28 | **58.00** | 0.47 | 14.89 | **59.50** | 0.49 | 0.24 | **66.00** | 0.48 | 6.77 | **13.00** | 0.48 | 5.46 |

## 3.2 SINGLE-PROPERTY OPTIMIZATION

Table 1 presents the performance across five single-property optimization tasks. POLO achieves the highest success rates on all tasks, averaging **84.3%** compared to **36.4%** for the best baseline.

**Instance-Specific vs. Generalizable Optimization.** Traditional methods like Graph-GA and Reinvent 4 require retraining from scratch for each new lead molecule, which is computationally prohibitive in practice. In contrast, POLO learns a generalizable optimization policy. It is trained once on a representative set of molecules for a given task and can then be directly applied to new, unseen molecules without further training. Despite this more challenging setting, POLO substantially outperforms all online methods. For example, Graph-GA, the strongest online baseline, achieves moderate success on QED (59.5%) and plogP (61.5%) but struggles with bioactivity targets (2.0% on JNK3). In comparison, POLO maintains strong performance across all properties.

**Impact of Domain-Specific Training.** General-purpose LLMs, despite their large scale, achieve limited success in lead optimziation. For example, Qwen2.5-7B and Llama3.1-8B achieve average success rates of only 20.4% and 23.5%, respectively. Task specific fine tuning provides a noticeable improvement, as demonstrated by GeLLM$^3$O, which raises the average success rate to 36.4%. In contrast, POLO's achieves an average success rate of **84.3%**, representing a substantial margin over existing LLMs. This result indicates that combining instruction tuning with reinforcement learning unlocks capabilities beyond supervised learning alone. The PGPO framework enables the model to learn effectively from exploration rather than simply mimicking training demonstrations.

**Impact of Task Complexity.** The performance gap between POLO and the baselines varies with task difficulty. For simpler physicochemical properties like QED and plogP, the margin of improvment is modest. In contrast, for challenging bioactivity targets which require precise navigation of the structure-activity relationship, POLO's advantage becomes substantial. This distinction is most pronounced in the complex bioactivity tasks, where POLO achieves a success rate of 81.0% on JNK3 versus 2.0% for the best baseline, and 97.0% on DRD2 versus 49.0%. This pattern of larger improvements on more difficult tasks validates the effectiveness of multi-turn optimization with preference learning for navigating complex optimization landscapes.

## 3.3 MULTI-PROPERTY OPTIMIZATION

Table 2 presents results on multi-property optimization tasks requiring simultaneous improvement of multiple, often competing objectives. Such scenarios closely mirror real drug discovery, where successful leads must satisfy diverse criteria. POLO maintains strong performance with an average success rate of **50.3%** across all multi-property tasks, compared to 24.9% for the best baseline (Reinvent 4). The most challenging three-property task (DRD2 +QED +plogP) highlights the exponential complexity of balancing bioactivity with physicochemical properties, where POLO achieves **13.0%** success rate while most baselines fail completely. These results indicate that multi-turn optimization with PGPO is particularly effective for multi-property tasks. The iterative refinement enables systematic exploration of the solution space, while preference learning guides the agent toward well-balanced compromises between competing objectives.

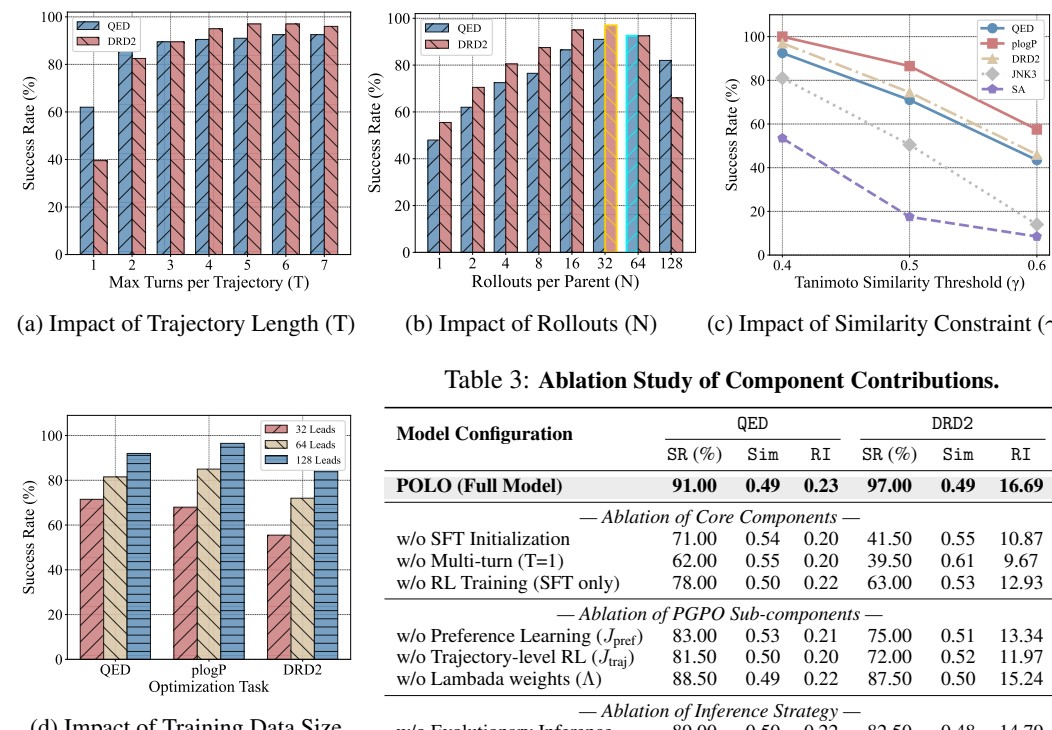

(a) Impact of Trajectory Length (T)    (b) Impact of Rollouts (N)    (c) Impact of Similarity Constraint ($\gamma$)

Table 3: **Ablation Study of Component Contributions.**

| Model Configuration | QED | | | DRD2 | | |
|---|---|---|---|---|---|---|
| | SR (%) | Sim | RI | SR (%) | Sim | RI |
| **POLO (Full Model)** | **91.00** | **0.49** | **0.23** | **97.00** | **0.49** | **16.69** |
| *— Ablation of Core Components —* | | | | | | |
| w/o SFT Initialization | 71.00 | 0.54 | 0.20 | 41.50 | 0.55 | 10.87 |
| w/o Multi-turn (T=1) | 62.00 | 0.55 | 0.20 | 39.50 | 0.61 | 9.67 |
| w/o RL Training (SFT only) | 78.00 | 0.50 | 0.22 | 63.00 | 0.53 | 12.93 |
| *— Ablation of PGPO Sub-components —* | | | | | | |
| w/o Preference Learning ($J_{\text{pref}}$) | 83.00 | 0.53 | 0.21 | 75.00 | 0.51 | 13.34 |
| w/o Trajectory-level RL ($J_{\text{traj}}$) | 81.50 | 0.50 | 0.20 | 72.00 | 0.52 | 11.97 |
| w/o Lambda weights ($\Lambda$) | 88.50 | 0.49 | 0.22 | 87.50 | 0.50 | 15.24 |
| *— Ablation of Inference Strategy —* | | | | | | |
| w/o Evolutionary Inference | 89.00 | 0.50 | 0.22 | 82.50 | 0.48 | 14.79 |

(d) Impact of Training Data Size

Figure 4: **Analysis of key hyperparameters. (a)** Multi-turn optimization shows clear benefits, with diminishing returns as T increases. **(b)** Increasing N initially improves performance through diverse exploration, but excessive parallelization limits evolutionary generations. Optimal performance is highlighted. **(c)** Stricter similarity constraints uniformly reduce performance. **(d)** POLO demonstrates strong data efficiency.

### 3.4 ANALYSIS OF KEY HYPERPARAMETERS

Figure 4 presents ablation analyses on critical hyperparameters to evaluate their impact on performance. **(1) Importance of multi-turn optimization.** As shown in Figure 3a, allowing the model to optimize over multiple turns yields substantial improvements compared to single-turn baselines. Notably, the performance gains saturate when the number of turns reaches approximately five to seven, indicating that this range provides an effective balance between optimization quality and efficiency. **(2) Exploration-exploitation trade-off.** Figure 3b highlights a nontrivial trade-off when the evaluation budget is limited. Increasing the number of rollouts initially enhances performance by promoting diverse exploration, but beyond roughly 32 rollouts we observe a pronounced decline. This degradation demonstrates that excessive parallelization limits the depth of evolutionary refinement, trading depth for breadth at the expense of optimization quality. **(3) Robustness to similarity constraints.** Figure 3c shows that while stricter similarity thresholds reduce success rates across tasks, POLO maintains functional performance even at $\gamma = 0.5$, demonstrating robustness to moderate constraints. **(4) Data efficiency.** Figure 3d demonstrates strong data efficiency of POLO, achieving **71.5%** success rate with only 32 training molecules. This efficiency is crucial given the scarcity of high-quality training examples. The rapid performance gain stems from PGPO's ability to extract $O(T^2)$ preference signals from each trajectory, amplifying the utility of limited training examples.

### 3.5 ABLATION STUDY

We present a systematic ablation study to quantify the contribution of each component to the overall performance as shown in Table 3. **(1) Multi-turn optimization is foundational.** Restricting to single-turn optimization (T=1) results in a substantial performance degradation ranging from 29% to 57%, which confirms the importance of an iterative process. **(2) PGPO components are complementary.** Removing either preference learning ($J_{\text{pref}}$) or trajectory-level RL ($J_{\text{traj}}$) causes a performance reduction of 10% to 25%, validating the dual-level design where trajectory signals provide strategic guidance while preferences offer fine-grained molecular comparisons. **(3) Chemical priors enable effective learning.** Without SFT initialization, DRD2 performance drops 55%, demonstrating that pre-training on valid chemical modifications is essential. **(4) Lambda weights and evolutionary inference amplify performance.** Lambda weights contribute 3% to 10% of improvement by

prioritizing informative comparisons, while evolutionary inference contributes 2% to 15% gains, with larger benefits on challenging targets. Together, these components form a cohesive framework where each element reinforces the others, creating POLO's superior optimization capability.

## 4 RELATED WORK

### 4.1 COMPUTATIONAL METHODS FOR LEAD OPTIMIZATION

To overcome the inefficiency and high cost of wet-lab experiments, researchers propose various computational methods to explore the chemical space more efficiently during lead optimization (Gao et al., 2022). Early computational approaches to this problem includes Genetic Algorithms (Jensen, 2019), Bayesian optimization (Korovina et al., 2020), and traditional machine learning methods. More recently, Reinforcement Learning has emerged as a powerful paradigm for goal-directed optimization (Popova et al., 2018; Olivecrona et al., 2017), with models like REINVENT 4 (Loeffler et al., 2024) demonstrating mature capabilities by integrating various architectures with staged learning. The comprehensive PMO benchmark demonstrates that most of these traditional methods fail under realistic budgets and struggle to balance property improvements with maintaining high structural similarity to the original lead compound (Gao et al., 2022). These limitations necessitate new approaches for sample-efficient and similarity-preserving optimization.

### 4.2 LARGE LANGUAGE MODELS FOR MOLECULAR DESIGN

Early transformer-based models showcase the potential of treating molecules as sequences (Chithrananda et al., 2020; Irwin et al., 2022; Bagal et al., 2021; Pei et al., 2023), achieving strong performance on property prediction and reaction tasks. The emergence of chemistry-aware LLMs marks a paradigm shift, enabling more sophisticated molecular optimization approaches. DrugAssist (Ye et al., 2025) and GeLLM$^3$O (Dey et al., 2025) fine-tune LLaMA models on instruction datasets for interactive optimization. MOLLEO (Wang et al., 2024a) employs LLMs as mutation operators within evolutionary algorithms, while ChatDrug (Liu et al., 2024a) enables conversational design through retrieval-augmented generation. However, these methods process each molecule independently without learning from optimization trajectories, discarding valuable historical context after each evaluation. This motivates our multi-turn RL approach, which enables LLMs to accumulate and leverage experience across complete optimization trajectories.

### 4.3 MULTI-TURN REINFORCEMENT LEARNING AND PREFERENCE LEARNING

Multi-turn Reinforcement Learning (multi-turn RL) emerges as a powerful framework for sequential decision-making with LLMs (Du et al., 2023; Wang et al., 2024b). This approach is central to the broader paradigm of Agentic RL, which reframes LLMs from passive generators into autonomous agents that learn through dynamic interaction with the environment (Zhang et al., 2025). Methods such as RAGEN (Wang et al., 2025) highlight the effectiveness of trajectory-level optimization for long-horizon strategies, demonstrating how cumulative reasoning across steps can enhance performance. In parallel, Preference Learning revolutionizes LLM alignment by providing a mechanism for dense, fine-grained feedback. Approaches like DPO (Rafailov et al., 2023) learn directly from comparisons, treating the LLM as an implicit reward function. This paradigm rapidly evolves from simple pairwise comparisons (Ethayarajh et al., 2024; Guo et al., 2025) to listwise formulations that optimize over entire rankings (Liu et al., 2024b). While these methods achieve remarkable success in language-based applications, their potential remains largely unexplored in domains such as molecular optimization.

## 5 CONCLUSION

In this work, we present POLO, a novel RL framework that transforms general-purpose LLMs into sample-efficient molecular optimization specialists. Unlike traditional methods that treat molecular modifications as isolated events, POLO addresses the iterative nature of lead optimization by enabling the LLM to learn from complete optimization trajectories. Our core contribution, Preference-Guided Policy Optimization (PGPO), leverages both trajectory-level strategic signals and turn-level preference comparisons from every molecular evaluation. This dual-level approach directly addresses the fundamental bottleneck of sample efficiency in lead optimization, where each oracle call represents significant experimental cost. Experiments demonstrate that POLO achieves state-of-the-art performance across diverse molecular optimization tasks using only 500 oracle evaluations. By enabling LLMs to learn from complete optimization trajectories rather than isolated steps, POLO not only advances molecular lead optimization but also provides a blueprint for applying language models to other scientific discovery tasks where iterative refinement and sample efficiency are paramount.

## ETHICS STATEMENT

This paper does not involve human subjects, personally identifiable data, or sensitive applications. We do not foresee direct ethical risks. We follow the ICLR Code of Ethics and affirm that all aspects of this research comply with the principles of fairness, transparency, and integrity.

## REPRODUCIBILITY STATEMENT

We ensure reproducibility of our experiments by fully describing the model architectures, datasets, preprocessing steps, hyperparameters, and training details in the main text and appendix. Code and scripts for reproducing the results are provided at `https://anonymous.4open.science/r/POLO-6861/`.

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

## A   LLM USAGE

We used Large Language Models (ChatGPT/Claude/Gemini) exclusively for grammatical correction in this manuscript. The LLMs played no role in research ideation, methodology, or scientific content generation. All technical contributions and scientific insights are original work by the authors.

## B   ENVIRONMENT IMPLEMENTATION DETAILS

### B.1   REWARD FUNCTION SPECIFICATION

The complete reward computation follows a hierarchical evaluation process that ensures validity before assessing property improvements:

---

**Algorithm 1** Reward Computation

---

1: **Input:** Current molecule $m_t$, Generated molecule $m_{t+1}$, Target property $F$
2: **Output:** Reward $r_t$, Success status
3: $m_{t+1}^{\text{parsed}} \leftarrow \text{ParseSMILES}(m_{t+1})$
4: **if** $m_{t+1}^{\text{parsed}} = \text{None}$ **then**                                                    ▷ Invalid SMILES
5:     **return** $r_t = -0.5$, success = False
6: **end if**
7: $m_t^{\text{canonical}} \leftarrow \text{Canonicalize}(m_t)$
8: $m_{t+1}^{\text{canonical}} \leftarrow \text{Canonicalize}(m_{t+1}^{\text{parsed}})$
9: **if** $m_{t+1}^{\text{canonical}} = m_t^{\text{canonical}}$ **then**                                    ▷ No modification
10:     **return** $r_t = -0.3$, success = False
11: **end if**
12: sim $\leftarrow \text{TanimotoSimilarity}(m_0, m_{t+1}^{\text{canonical}})$
13: **if** sim $< \gamma$ **then**                                                    ▷ Similarity threshold violation
14:     **return** $r_t = -2(\gamma - \text{sim})$, success = False
15: **end if**
16: $\Delta F \leftarrow F(m_{t+1}^{\text{canonical}}) - F(m_t^{\text{canonical}})$
17: $r_t \leftarrow \begin{cases} 5|\Delta F| & \text{if improvement } (\text{sgn}(w_F) \cdot \Delta F > 0) \\ -|\Delta F| & \text{otherwise} \end{cases}$
18: success $\leftarrow (\text{sgn}(w_F) \cdot \Delta F > 0)$
19: **return** $r_t$, success

---

Table 4: Reward structure for different modification outcomes.

| Condition | Reward | Success Status |
|---|---|---|
| Invalid SMILES | $-0.5$ | Failed |
| No modification (identical molecule) | $-0.3$ | Failed |
| Similarity $< \gamma$ | $-2(\gamma - \text{sim})$ | Failed |
| Property degradation | $-|\Delta F|$ | Failed |
| Property improvement | $5 \times |\Delta F|$ | Success |

Table 4 summarizes the reward values for different modification outcomes. The reward structure prioritizes valid chemical modifications while encouraging exploration through asymmetric rewards — improvements receive $5\times$ amplification while degradations incur smaller penalties, preventing the agent from becoming overly conservative.

### B.2   ROLLBACK MECHANISM

To prevent the agent from getting stuck after failed modifications, the environment tracks the best-scoring molecule $m_{\text{best}}$ encountered during each trajectory. After any failed modification (negative reward), if the best molecule differs from and outperforms the current one, the environment reverts to $m_{\text{best}}$ and provides feedback suggesting alternative strategies. This ensures monotonic improvement in tracked performance while enabling efficient exploration from proven successful states.

---

**Algorithm 2** The POLO Training Framework

---

**Require:** Stage-1 fine-tuned model $\pi_\theta$; Lead molecules $\mathcal{M}_0 = \{m_0^{(1)}, \ldots, m_0^{(M)}\}$; Training itera-
    tions $I$; Rollouts per lead $N$; Trajectory horizon $T$; Variance filter ratio $k = 0.5$; Score filter
    ratio $p = 0.75$; Preference loss weight $\lambda_{\text{pref}}$.
**Ensure:** Optimized policy $\pi_{\theta*}$.
  1: **for** iteration $i = 1$ to $I$ **do**
  2:     $\mathcal{D}_{\text{batch}} \leftarrow \emptyset$
                  *— Phase 1: Trajectory Sampling —*
  3:     **for** each lead molecule $m_0^{(j)} \in \mathcal{M}_0$ **do**
  4:         **for** rollout $n = 1$ to $N$ **do**
  5:             Initialize $s_0$ with task instruction and $m_0^{(j)}$
  6:             Generate trajectory $\tau_n = (s_0, a_0, r_0, \ldots, a_{T-1}, r_{T-1}, s_T)$ using $\pi_\theta$
  7:             $\mathcal{D}_{\text{batch}} \leftarrow \mathcal{D}_{\text{batch}} \cup \{\tau_n\}$
  8:         **end for**
  9:     **end for**
              *— Phase 2: Two-Stage Trajectory Filtering —*
10:     Partition $\mathcal{D}_{\text{batch}}$ into groups $\{G_1, \ldots, G_M\}$ by lead molecule
11:                              ▷ Stage 1: Select high-variance groups (top 50%)
12:     Compute std for each group: $\sigma_j = \text{Std}_{\tau \in G_j}(R(\tau))$ where $R(\tau) = \sum_t r_t$
13:     $\mathcal{G}_{\text{filtered}} \leftarrow \{G_j \mid \sigma_j \geq \text{median}(\{\sigma_1, \ldots, \sigma_M\})\}$
14:                           ▷ Stage 2: Select top trajectories within groups (top 75%)
15:     $\mathcal{D}_{\text{filtered}} \leftarrow \emptyset$
16:     **for** each group $G_j \in \mathcal{G}_{\text{filtered}}$ **do**
17:         $\mathcal{T}_{\text{top}} \leftarrow \{\tau \in G_j \mid R(\tau) \geq \text{percentile}_{25}(\{R(\tau')\}_{\tau' \in G_j})\}$
18:         $\mathcal{D}_{\text{filtered}} \leftarrow \mathcal{D}_{\text{filtered}} \cup \mathcal{T}_{\text{top}}$
19:     **end for**
              *— Phase 3: Dual-Level Policy Optimization —*
20:     **for** each $\tau \in \mathcal{D}_{\text{filtered}}$ **do**
21:         Compute advantages $\hat{A}_t$ using GAE for trajectory-level learning
22:         Extract turn pairs and compute Lambda weights for preference learning
23:     **end for**
24:     Update $\theta$ by optimizing: $J_{\text{traj}}(\theta) + \lambda_{\text{pref}} J_{\text{pref}}(\theta)$
25: **end for**
26: **return** $\pi_{\theta*}$

---

## B.3 TRAINING ALGORITHM

Algorithm 2 formalizes the complete POLO training procedure, integrating trajectory sampling, filtering, and dual-level optimization.

The three-phase design enables efficient learning from expensive oracle evaluations. Phase 1 generates diverse optimization trajectories across multiple leads. Phase 2 applies our two-stage filtering strategy (detailed in Appendix C) to reduce the dataset to approximately 37.5% of trajectories. Phase 3 applies our dual-level optimization, combining trajectory-level PPO for strategic learning with turn-level preference learning for fine-grained molecular improvements. This selective training accelerates convergence while focusing on the most informative experiences.

## B.4 IMPLEMENTATION NOTES

Episodes terminate when: (1) reaching maximum steps $T = 5$, (2) agent outputs [DONE], or (3) achieving task-specific success criteria. Tanimoto similarity is computed using Morgan fingerprints (radius=2, 2048 bits). Default environment settings include similarity threshold $\gamma = 0.4$, penalty weight $\lambda = 1.0$, and reward amplification factor of 5. For comprehensive hyperparameter settings and baseline configurations, see Appendix G.

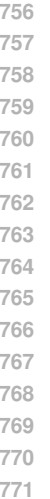
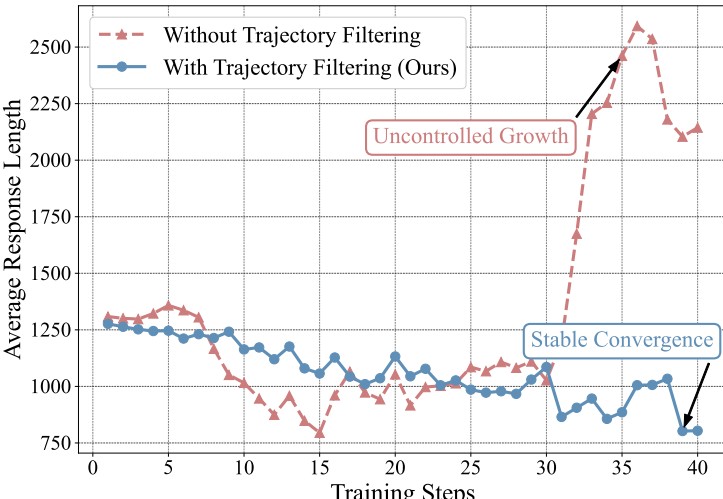

Figure 5: **Trajectory filtering prevents training collapse.** Without filtering (red), response length explodes to 2,500 tokens after step 35, indicating model collapse into repetitive generation. With filtering (blue), training remains stable at 1,000 tokens throughout.

## C  TRAJECTORY FILTERING FOR TRAINING STABILITY

To ensure stable training and prevent model collapse, we employ a two-stage filtering strategy that identifies and prioritizes high-quality training data from collected trajectories. As shown in Figure 5, this filtering is crucial for preventing catastrophic degradation during training.

### C.1  MOTIVATION: PREVENTING MODEL COLLAPSE

Our empirical observations align with recent findings in multi-turn RL (Wang et al., 2025) that training on low-quality trajectories can lead to irreversible model degradation. Without filtering, the model exhibits unstable training dynamics: the average response length suddenly explodes after approximately 35 training steps, indicating the model has collapsed into generating repetitive, meaningless tokens (Figure 5, red line). This collapse is characterized by:

- **Uncontrolled response growth:** Response length increases from  1000 to >2500 tokens

- **Semantic degradation:** Output becomes repetitive and non-informative

- **Irreversible damage:** Once collapsed, the model cannot recover even with continued training

### C.2  TWO-STAGE FILTERING PROCESS

Our filtering strategy ensures training stability by selecting only high-quality, informative trajectories:

**Stage 1: Group-Level Selection by Diversity** For each lead molecule $m_i$, we collect multiple trajectory rollouts forming a group $G_i$. Groups with high reward variance indicate challenging optimization problems that provide rich learning signals. We select groups where:

$$\text{Select}(G_i) = \mathbf{1}\left[\text{Std}_{\tau \in G_i}(R(\tau)) \geq \text{median}\left(\{\text{Std}(G_j)\}_j\right)\right]$$

This retains 50% of groups with highest variance, ensuring diverse optimization challenges while filtering out trivial or degenerate trajectories.

**Stage 2: Instance-Level Selection by Quality** From each selected group, we keep the top 75% of trajectories by cumulative reward:

$$\text{Keep}(\tau|G_i \text{ selected}) = \mathbf{1}\left[R(\tau) \geq \text{percentile}_{25}\left(\{R(\tau') : \tau' \in G_i\}\right)\right]$$

This threshold (75%) is carefully chosen: it maintains sufficient diversity for preference learning while excluding completely failed attempts that could destabilize training.

## C.3 EMPIRICAL VALIDATION AND IMPLICATIONS

Figure 5 demonstrates the critical importance of this filtering strategy. With our two-stage filtering (blue line), training exhibits stable convergence with response length around 800-1000 tokens, while without filtering (red line), the model catastrophically collapses after 35 steps with response length exploding to >2,500 tokens. The filtering parameters (50% group selection, 75% trajectory retention) were determined through extensive experimentation—more aggressive filtering (e.g., 30%/50%) led to insufficient training data, while less filtering (e.g., 70%/90%) increased instability risk.

This filtering strategy is particularly critical for PGPO's dual-level learning mechanism. Low-quality trajectories not only provide poor trajectory-level signals but also generate misleading preference pairs. By ensuring training data quality through filtering, we achieve:

- **Training stability:** Prevents model collapse into repetitive, meaningless generation

- **Robust learning signals:** High-variance groups provide diverse yet valid optimization strategies for trajectory learning, while quality filtering ensures preference pairs represent genuine improvements

- **Computational efficiency:** Training on only 37.5% of trajectories reduces computational cost while maintaining or improving final performance

## D INFERENCE-TIME EVOLUTIONARY REFINEMENT

While PGPO training equips our agent with effective optimization strategies, standard inference typically runs independent rollouts where each trajectory is amnesiac to others' successes (Loeffler et al., 2024). To maximize performance without additional training, we deploy our agent within an evolutionary framework that transforms it into an intelligent mutation operator.

### D.1 KEY DESIGN CHOICES

The evolutionary refinement strategy leverages our trained POLO agent as an intelligent mutation operator within a population-based search framework. Unlike traditional evolutionary algorithms that rely on random mutations, our approach uses the agent's learned optimization strategies to propose chemically meaningful modifications.

**Offspring Generation.** Each parent molecule spawns $N = 32$ independent optimization trajectories, with each trajectory producing up to $T = 5$ candidate molecules. This multi-trajectory approach increases diversity while the learned policy ensures modifications are more likely to be beneficial compared to random mutations.

**Elite Pool Dynamics.** The elite pool maintains the top $K = 5$ molecules discovered during optimization. New molecules enter the pool if they satisfy the similarity constraint ($\gamma_{\text{elite}} = 0.4$) and either the pool is not full or they outperform the current worst member. This mechanism ensures continuous improvement while preserving diversity.

**Budget Management.** With a limited oracle budget of $B = 500$ evaluations, the algorithm must balance exploration and exploitation. The uniform parent selection promotes diversity, while the learned mutation operator ensures efficient use of each evaluation. The algorithm typically converges within 5-10 generations, well before exhausting the budget.

**Oracle Caching.** Following standard benchmarking practice, we implement oracle caching to avoid redundant evaluations. Each unique molecule is evaluated at most once, with results stored for potential reuse. This is particularly important given our multi-trajectory approach, which may generate duplicate candidates.

### D.2 IMPLEMENTATION NOTES

The trained POLO agent uses a dynamic temperature scheduling strategy to balance exploration and exploitation across generations:

$$\tau_g = \min(\tau_{\text{base}} + (g - 1) \cdot \Delta\tau, \tau_{\text{max}}) \tag{7}$$

where $\tau_{\text{base}} = 0.9$ is the initial temperature, $\Delta\tau = 0.1$ is the increment per generation, and $\tau_{\text{max}} = 2.0$ caps the maximum temperature. This schedule starts with more deterministic generation to exploit learned patterns, then gradually increases randomness to encourage exploration as optimization progresses.

---

**Algorithm 3** POLO Inference via Evolutionary Refinement

---

**Require:** Trained POLO agent $\pi_{\theta*}$; Initial lead molecule $m_0$; Oracle budget $B = 500$; Generations $G = 10$; Elite pool capacity $K = 5$; Elite similarity threshold $\gamma_{\text{elite}} = 0.4$; Rollouts per parent $N = 32$; Rollout horizon $T = 5$.
**Ensure:** Final elite pool $\mathcal{E}_G$ containing optimized molecules.
 1: Initialize elite pool $\mathcal{E}_0 \leftarrow \{m_0\}$
 2: Evaluate $F(m_0)$ via oracle
 3: Initialize oracle call counter $b \leftarrow 1$
 4: **for** generation $g = 1$ to $G$ **do**
 5:      **if** $b \geq B$ **then break**                     ▷ Budget exhausted
 6:      **end if**

                               *— Parent Selection —*

 7:      Sample parent $m_p \sim \text{Uniform}(\mathcal{E}_{g-1})$

                               *— Offspring Generation —*

 8:      $\mathcal{M}_{\text{candidates}} \leftarrow \emptyset$
 9:      **for** rollout $i = 1$ to $N$ **do**
10:          Initialize conversation state $s_0$ with $m_p$ as starting molecule
11:          **for** turn $t = 1$ to $T$ **do**
12:              Generate molecule $m_t$ using $\pi_{\theta*}(a_t|s_{t-1})$
13:              **if** $\text{sim}(m_0, m_t) \geq \gamma_{\text{elite}}$ **then**
14:                  $\mathcal{M}_{\text{candidates}} \leftarrow \mathcal{M}_{\text{candidates}} \cup \{m_t\}$
15:              **end if**
16:              Update state $s_t$ with $m_t$ and oracle feedback
17:              **if** agent outputs [DONE] or $b \geq B$ **then break**
18:              **end if**
19:          **end for**
20:      **end for**

                               *— Elite Pool Update —*

21:      **for** each unique $m \in \mathcal{M}_{\text{candidates}}$ **do**
22:          **if** $b < B$ **then**
23:              Evaluate $F(m)$ via oracle; $b \leftarrow b + 1$
24:              **if** $\text{sim}(m_0, m) \geq \gamma_{\text{elite}}$ **then**
25:                  Update $\mathcal{E}_{g-1}$ by adding $m$ if better than worst member or pool not full
26:              **end if**
27:          **end if**
28:      **end for**
29:      $\mathcal{E}_g \leftarrow \mathcal{E}_{g-1}$
30: **end for**
31: **return** $\mathcal{E}_G$

---

For multi-property optimization, we use difficulty-weighted scoring to fairly compare molecules with different improvement profiles. The fitness function is:

$$F(m) = \sum_i w_i \cdot \Delta F_i(m) \tag{8}$$

where $\Delta F_i(m) = F_i(m) - F_i(m_0)$ is the absolute improvement for property $i$, and weights $w_i$ are inversely scaled to typical improvement magnitudes: $w_{\text{JNK3}} = 12$ (hardest, typical $\Delta = 0.05\text{-}0.1$), $w_{\text{QED}} = 10$, $w_{\text{DRD2}} = 4$, $w_{\text{SA}} = 2$, and $w_{\text{logP}} = 1$ (easiest, typical $\Delta = 2\text{-}3$). This weighting ensures that improvements in challenging properties like JNK3 are valued appropriately against easier properties like logP.

Episodes terminate early if the agent outputs [DONE] or achieves task-specific success criteria. Oracle caching prevents redundant evaluations — each unique molecule is evaluated at most once, with results stored for potential reuse. Complete implementation details and hyperparameter sensitivity analysis are provided in our code release.

# E  MolOptIns: Teaching LLMs Controlled Molecular Modification

A fundamental challenge in leveraging LLMs for lead optimization is their difficulty in performing precise, similarity-preserving molecular edits. While powerful in generation, existing models often struggle with the controlled, local modifications required to refine lead compounds. Prior work shows that even specialized models produce molecules with low resemblance to starting points — for instance, BioT5 and GPT-4 achieve only 0.173 and 0.165 Tanimoto similarity respectively (Wang et al., 2024a). To address this deficiency, we constructed **MolOptIns**, an instruction-tuning dataset specifically designed to teach LLMs controlled molecular modification.

## E.1  Dataset Construction

Following the methodology of GeLLM³O (Dey et al., 2025), we built MolOptIns from the molecule pair dataset of Chen et al. (Chen et al., 2021). Each pair $(M_x, M_y)$ represents a single-fragment modification, filtered to retain only pairs with Tanimoto similarity $\geq 0.6$. This ensures every example demonstrates a successful, structure-preserving optimization step. For each pair, we computed five key properties (QED, plogP, JNK3, DRD2, SA) and categorized them into optimization tasks following GeLLM³O's task definitions. Pairs were selected only if they showed meaningful improvement for their corresponding task while maintaining high similarity. Each filtered pair is converted to instruction-response format using the unified prompt template shown below. The final dataset contains 500K high-quality examples.

---

**OUnified Prompt Template**

```
You are an expert medicinal chemist specializing in molecular optimization. You
understand how structural modifications affect key molecular properties including
drug-likeness, lipophilicity, synthetic accessibility, and target inhibition
activities.
Your task is to modify the given molecule to adjust the specified molecular
properties while keeping structural changes as minimal as possible. The modified
molecule should maintain a structural similarity of at least 0.4 with the original
molecule.
Input molecule: <SMILES> {input_smiles} </SMILES>
Requested modifications: {property_description}
Please provide the optimized molecule in SMILES format, wrapped in <SMILES>
</SMILES> tags.
```

---

The {property_description} field is populated based on the task:

- QED: increase drug-likeness (QED)
- LogP: increase lipophilicity (LogP)
- JNK3/DRD2: increase inhibition probability for the specified target
- SA: decrease synthetic accessibility score (lower is better)
- Multi-property: combine the above objectives with "and" conjunctions

## E.2  Prompt Template Usage

The unified prompt template serves multiple critical roles throughout POLO's pipeline:

- **Stage 1 - Supervised Fine-tuning:** Used to create instruction-response pairs for the 500K MolOptIns dataset

- **Stage 2 - PGPO Training:** Serves as the initial state $s_0$ in the MDP formulation, establishing the task context for multi-turn optimization

- **Inference:** Initializes each rollout trajectory during evolutionary refinement

This consistent prompting strategy ensures continuity across training stages and enables the model to leverage its learned instruction-following capabilities throughout the optimization process.

### E.3 FINE-TUNING PROCEDURE

We fine-tune Qwen2.5-1.5B-Instruct on MolOptIns using LoRA (Hu et al., 2022) (rank=16, alpha=32) for parameter-efficient training, running for 10 epochs. The fine-tuned model serves as both the initialization for $\pi_\theta$ and the reference policy $\pi_{\text{ref}}$ in our PGPO framework.

### E.4 ROLE IN PGPO FRAMEWORK

The model fine-tuned on MolOptIns serves dual critical roles:

- **Initialization:** Provides the starting policy $\pi_\theta$ with foundational chemical modification capabilities

- **Regularization:** Acts as the reference policy $\pi_{\text{ref}}$ in preference learning, preventing deviation into chemically invalid regions

This similarity-aware fine-tuning bridges the gap between general language models and specialized molecular optimizers, ensuring that even before PGPO training, the model can generate chemically valid and structurally faithful modifications.

## F BASELINE METHODS AND EXPERIMENTAL CONFIGURATION

### F.1 BASELINE DESCRIPTIONS

We evaluate POLO against 10 baseline methods spanning four categories, each representing different approaches to molecular optimization.

#### F.1.1 TRADITIONAL METHODS

- **Graph-GA (Graph-based Genetic Algorithm)** (Jensen, 2019): A classic evolutionary algorithm that evolves a population of molecular graphs. It uses domain-specific crossover and mutation operators designed for molecular structures to optimize the given objective function while maintaining chemical validity.

- **QMO (Query-based Molecule Optimization)** (Hoffman et al., 2022): A black-box optimization framework operating in the latent space of a pre-trained molecular autoencoder. It employs zeroth-order optimization algorithms that estimate gradients through random perturbations, enabling optimization without explicit gradient information.

- **Reinvent 4** (Loeffler et al., 2024): A reinforcement learning approach using an RNN-based agent trained via policy gradient methods. The agent learns to generate SMILES strings that maximize a predefined scoring function through on-policy training with reward shaping.

#### F.1.2 GENERAL-PURPOSE LLMS

- **Qwen2.5-7B** (Yang et al., 2025): A state-of-the-art foundation model with strong reasoning capabilities, tested in zero-shot molecular optimization.

- **Llama3.1-8B** (Dubey et al., 2024): Meta's instruction-tuned model, evaluated for its ability to understand and execute molecular modification tasks.

#### F.1.3 TASK-SPECIFIC LLMS

- **MOLLEO** (Wang et al., 2024a): A hybrid approach combining Graph-GA's evolutionary framework with LLM-powered mutation operators, using BioT5 or GPT-4 to generate chemically meaningful modifications.

- **LlaSMol** (Yu et al., 2024): A series of chemistry-specialized LLMs created by fine-tuning various base models (e.g., Mistral) on **SMolInstruct**, a large-scale, high-quality instruction dataset covering 14 diverse chemistry tasks. It represents a strong baseline for general chemical reasoning and instruction-following capabilities. In the experiments, we use the Mistral-7b version.

- **ChemLLM** (Zhang et al., 2024): A chemistry-specialized LLM fine-tuned on millions of chemical Q&A pairs, molecular property predictions, and reaction data.

- **PEIT-LLM** (Lin et al., 2024): A framework that first pre-trains a multi-modal generator (PEIT-GEN) to align textual, structural (SMILES), and biochemical property data. This generator is then used as a data synthesizer to create a large-scale instruction dataset, which is subsequently used to

fine-tune a general-purpose LLM (in this case, LLaMA3.1-8B) for various multi-task molecular generation challenges.

- **GeLLM$^3$O** (Dey et al., 2025): A state-of-the-art model for multi-property optimization, instruction-tuned on the MuMOInstruct dataset containing successful molecular optimization trajectories.

### F.2 IMPLEMENTATION DETAILS

**Traditional ML Methods.** We use official implementations with default hyperparameters. Each method's optimization loop is modified to track oracle calls.

**LLM-Based Methods.** All LLM baselines follow a unified evaluation protocol: (1) prompt the model with the optimization task, (2) extract SMILES from the response, (3) validate and canonicalize the molecule, (4) evaluate if valid, (5) repeat until 500 valid molecules are evaluated. The best molecule satisfying all constraints is reported. All LLMs use temperature $\tau = 0.9$ for generation. For fair comparison, all LLM baselines receive the identical prompting template described in Appendix E.

## G HYPERPARAMETER SETTINGS

### G.1 POLO TRAINING CONFIGURATION

Table 5 summarizes the key hyperparameters used in POLO training.

Table 5: POLO training hyperparameters.

| Category | Parameter | Value |
|---|---|---|
| Model | Base model | Qwen2.5-1.5B-Instruct |
| | LoRA rank | 16 |
| | LoRA alpha | 32 |
| | Max sequence length | 4096 |
| Training | Training steps | 100 |
| | Micro batch size per GPU | 2 |
| | PPO mini batch size | 32 |
| | Learning rate | $5 \times 10^{-5}$ |
| | GPUs | $2\times$ H100 |
| Environment | Max turns per trajectory | 5 |
| | Training molecules | 128 |
| | Rollouts per molecule | 16 |
| | Similarity threshold $\gamma$ | 0.4 |
| PPO | Discount factor $\gamma$ | 0.99 |
| | GAE lambda $\lambda$ | 0.95 |
| | Clip ratio $\epsilon$ | 0.2 |
| Preference Learning | Preference loss weight $\lambda_{\text{pref}}$ | 0.3 |
| | Temperature $\tau_{\text{temp}}$ | 0.9 |
| | Max intra-trajectory pairs | 6 |
| Trajectory Filtering | Variance filter ratio | 0.5 |
| | Score filter ratio | 0.75 |
| | Filter metric | Std. deviation |

### G.2 INFERENCE CONFIGURATION

Table 6 summarizes the hyperparameters used during inference and evaluation.

## H EVALUATION METRICS DETAILS

### H.1 SUCCESS RATE (SR)

Success Rate measures the percentage of lead molecules successfully optimized. A successful optimization must satisfy both the similarity constraint ($\text{sim}(m, m') \geq 0.4$) and task-specific property criteria:

where $\Delta F = F(m') - F(m)$ represents the property change. For multi-property tasks, ALL target properties must meet their respective improvement thresholds.

Table 6: POLO inference hyperparameters.

| Category | Parameter | Value |
|---|---|---|
| Budget & Constraints | Oracle budget $B$ | 500 |
| | Similarity threshold $\gamma$ | 0.4 |
| | Max generations $G$ | 10 |
| Evolutionary Strategy | Elite pool size $K$ | 5 |
| | Rollouts per parent $N$ | 32 |
| | Max turns per rollout $T$ | 5 |
| Temperature Schedule | Initial temperature $\tau_{\text{base}}$ | 0.9 |
| | Temperature increment $\Delta\tau$ | 0.1 |
| | Max temperature $\tau_{\text{max}}$ | 2.0 |

Table 7: Success criteria for single-property and multi-property optimization tasks

| Property | Single-Property Task | | Multi-Property Task | |
|---|---|---|---|---|
| | Criterion | Threshold | Criterion | Threshold |
| QED | $F_{\text{QED}}(m') \geq$ | 0.9 | $\Delta F_{\text{QED}} \geq$ | 0.1 |
| plogP | $F_{\text{plogP}}(m') \geq$ | 2.0 | $\Delta F_{\text{plogP}} \geq$ | 1.0 |
| JNK3 | $F_{\text{JNK3}}(m') \geq$ | 0.4 | $\Delta F_{\text{JNK3}} \geq$ | 0.1 |
| DRD2 | $F_{\text{DRD2}}(m') \geq$ | 0.8 | $\Delta F_{\text{DRD2}} \geq$ | 0.5 |
| SA | $F_{\text{SA}}(m') \leq$ | 2.5 | $-\Delta F_{\text{SA}} \geq$ | 0.5 |

## H.2 SIMILARITY (Sim)

The average Tanimoto similarity is computed across all test molecules:

$$\text{Sim} = \frac{1}{N} \sum_{i=1}^{N} \text{sim}(m_i, m_i') \qquad (9)$$

where $N = 200$, $m_i$ is the $i$-th lead molecule, and $m_i'$ is its best optimized variant. For failed optimizations where no valid molecule meeting the constraints is found, we set $m_i' = m_i$, yielding $\text{sim}(m_i, m_i') = 1.0$.

This computation choice ensures fair comparison across methods with varying success rates. Methods with low success rates (many failures) will show artificially high similarity scores, as failures contribute the maximal similarity of 1.0. This reveals an important trade-off: baseline methods often fail to explore the chemical space effectively under the given constraints, defaulting to the original molecule, while successful methods like POLO demonstrate the ability to find diverse yet valid optimizations within the similarity budget.

## H.3 RELATIVE IMPROVEMENT (RI)

The average percentage improvement across target properties:

$$\text{RI} = \frac{1}{n} \sum_{j=1}^{n} \text{sgn}(w_j) \cdot \frac{F_j(m') - F_j(m)}{|F_j(m)|} \qquad (10)$$

where $n$ is the number of properties, $\text{sgn}(w_j) = +1$ for properties to maximize (QED, plogP, JNK3, DRD2) and $-1$ for properties to minimize (SA). The absolute value $|F_j(m)|$ in the denominator handles properties that can take negative values (e.g., plogP). Failed optimizations yield RI = 0.

## I COMPUTATIONAL RESOURCES AND COST ANALYSIS

We provide a comprehensive analysis of computational costs across different optimization paradigms. The key distinction lies in how costs are amortized: traditional methods require retraining for each task, while POLO's training cost can be shared across multiple optimization tasks.

Table 8: **Computational Cost Comparison.** GPU hours on $2\times$ H100 GPUs. †One-time cost amortizable across all tasks. *Reinvent 4 requires retraining for each new lead molecule.

| Method | Model Size | One-time Training | Per-task Training | Per-task Inference | Total per Task |
|---|---|---|---|---|---|
| Reinvent 4* | – | – | 14h (online) | – | 14h |
| GeLLM³O | 7B | 48h† (SFT) | – | 1h | 1h |
| **POLO** | **1.5B** | **10h† (SFT)** | **3h (PGPO)** | **2h** | **5h** |

POLO's training consists of two stages: (1) one-time SFT on MolOptIns (10 hours on $2\times$ H100 GPUs), reusable across all tasks, and (2) task-specific PGPO training (3 hours). Inference takes approximately 2 hours for 200 test molecules with 500 oracle evaluations.

Compared to existing approaches, POLO's computational requirements are practical: Reinvent 4 requires 14 hours of online training for each optimization instance, while POLO needs only 3 hours of task-specific training after the one-time SFT. Although our multi-turn approach generates T=5 molecules sequentially per trajectory, efficient batching keeps inference time reasonable at 2 hours per task. The use of a smaller 1.5B parameter model (compared to 7-8B in other LLM baselines) also enables deployment on more accessible hardware while achieving superior performance.

## J ANALYSIS OF PGPO'S SAMPLE EFFICIENCY MECHANISM

We analyze the mechanism underlying PGPO's empirical sample efficiency. The key insight is that PGPO extracts significantly more learning signals from the same number of expensive oracle evaluations.

### J.1 SIGNAL AMPLIFICATION THROUGH DUAL-LEVEL LEARNING

The fundamental difference between standard RL and PGPO lies in the density of learning signals extracted from collected trajectories.

For standard RL methods using $N$ trajectories, the number of primary learning signals is:

$$S_{\text{RL}} = N \tag{11}$$

Each trajectory contributes exactly one scalar reward signal that must be credited across all $T$ decisions in that trajectory.

In contrast, PGPO combines trajectory-level rewards with turn-level preference pairs. For each trajectory of length $T$, we can construct up to $\binom{T}{2}$ pairwise comparisons between different turns. Thus, the total number of learning signals becomes:

$$S_{\text{PGPO}} = \underbrace{N}_{\text{trajectory-level}} + \underbrace{\sum_{i=1}^{N} k_i}_{\text{turn-level}} \approx N + N \cdot \frac{T(T-1)}{2} = O(NT^2) \tag{12}$$

This represents a quadratic amplification in the number of learning signals:

- **Standard RL:** $O(N)$ signals from $N$ trajectory rewards
- **PGPO:** $O(NT^2)$ signals from both trajectory rewards and preference pairs

### J.2 PRACTICAL BENEFITS OF DENSE LEARNING SIGNALS

The increased signal density provides several practical advantages:

- **Direct Credit Assignment:** Each preference pair $(t_1, t_2)$ where $r_{t_2} > r_{t_1}$ provides a direct signal that action $a_{t_2}$ is preferred over $a_{t_1}$ in similar contexts. This bypasses the challenging problem of distributing a single trajectory reward across $T$ sequential decisions.
- **Relative Feedback Robustness:** Learning from relative comparisons ("this modification is better than that one") is often more stable than learning from absolute trajectory scores, as relative rankings are less sensitive to environmental noise and exploration artifacts.

- **Efficient Use of Oracle Budget:** By generating $O(T^2)$ training signals per trajectory, PGPO maximizes the learning value extracted from each expensive oracle call. This is particularly crucial in drug discovery where each property evaluation represents significant computational or experimental cost.

**Remark.** In practice, we employ a selective strategy rather than using all $\binom{T}{2}$ possible pairs. Specifically, we select the top 75% of pairs ranked by their reward difference $|r_{t_2} - r_{t_1}|$. This filtering serves dual purposes: (1) it accelerates training by reducing computational overhead from $O(T^2)$ to $O(0.75T^2)$, and (2) it improves signal quality by focusing on the most informative comparisons — pairs with larger reward differences provide clearer learning signals about relative action quality. Additionally, Lambda weighting (Eq. 6) further prioritizes pairs that are currently misranked, ensuring the model focuses on correcting its most significant errors. Despite using only a subset of possible pairs, the signal amplification remains substantial compared to standard RL's $O(N)$ signals, as demonstrated by our experimental results showing consistent improvements in sample efficiency across all tasks.

## K  OUT-OF-DISTRIBUTION GENERALIZATION

To comprehensively evaluate POLO's generalization capabilities, we conduct two sets of out-of-distribution (OOD) experiments: (1) robustness to different instruction phrasings, and (2) cross-task transfer ability. These experiments test whether POLO has learned generalizable optimization strategies or merely memorized training patterns.

### K.1  OUT-OF-DISTRIBUTION PROMPTING

We evaluate POLO's instruction generalization capabilities using three prompt styles that deviate significantly in tone and structure from the standard training template: a **Casual** conversational style, a highly structured **Technical** format, and an open-ended **Creative** instruction.

> **OOD Prompt Style 1: Casual**
>
> Hey! You're a chemist working on molecules. Take this molecule and make it better for {property_description}. Don't change it too much though - keep similarity above 0.4. Just give me the modified SMILES string.

> **OOD Prompt Style 2: Technical**
>
> OBJECTIVE: Perform molecular structure optimization targeting {property_description}. CONSTRAINTS: Maintain Tanimoto similarity coefficient $\geq$ 0.4. METHOD: Apply systematic structural modifications using computational chemistry principles. OUTPUT: Optimized molecular structure in SMILES notation.

> **OOD Prompt Style 3: Creative**
>
> Imagine you're designing the perfect molecule. Transform the given structure to {property_description} while keeping its molecular identity. Think creatively but stay within similarity bounds of 0.4. What would your ideal molecule look like?

Table 9 demonstrates remarkable robustness across all prompt variations. Performance retention exceeds 97% for both QED and DRD2 tasks, with Casual and Creative styles achieving nearly 99.5% retention. The minimal degradation indicates that POLO has learned the underlying optimization semantics rather than overfitting to specific phrasings. This robustness is crucial for practical deployment where users may express objectives in diverse natural language forms.

### K.2  CROSS-TASK TRANSFER PERFORMANCE

We investigate whether the optimization strategies learned by POLO are task-specific or if they represent a more generalizable chemical reasoning capability. To this end, we conduct a cross-task generalization experiment where agents trained on one property are evaluated on others.

Table 10 reveals selective transfer patterns that align with underlying chemical principles. The most notable finding is the exceptional JNK3 →QED transfer, achieving 76.0% success rate — only 15

Table 9: **Robustness to Out-of-Distribution (OOD) Prompts.** We evaluate POLO's instruction generalization using three unseen prompt styles. The model maintains high performance, demonstrating it has learned the underlying task rather than overfitting to a specific phrasing.

| Prompt Style | QED | | | DRD2 | | |
|---|---|---|---|---|---|---|
| | SR (%) | Sim | RI | SR (%) | Sim | RI |
| Original (Baseline) | **91.00** | 0.49 | 22.77 | **97.00** | 0.48 | 16.70 |
| Casual | 90.50 (99.50%) | 0.49 | 22.78 | 96.00 (99.00%) | 0.48 | 16.60 |
| Technical | 89.00 (97.80%) | 0.51 | 22.56 | 95.50 (98.50%) | 0.48 | 16.62 |
| Creative | 90.50 (99.50%) | 0.49 | 22.63 | 96.50 (99.50%) | 0.48 | 16.79 |

SR, Sim, and RI are reported. Values in parentheses under SR denote the performance retention relative to the Standard prompt. Best performance for each metric is in **bold**.

Table 10: **Cross-Task Generalization Performance of POLO Agents.** Specialized agents trained on one task are evaluated on other unseen tasks. Each block compares an agent's performance on its native task (in-domain) versus an unseen (OOD) task.

| Training Model | Evaluation Task | SR (%) | Sim | RI |
|---|---|---|---|---|
| Trained on QED | QED (*In-Domain*) | **91.00** | **0.49** | **22.77** |
| | plogP (*OOD*) | 13.50 | 0.75 | 0.08 |
| Trained on plogP | plogP (*In-Domain*) | **99.00** | **0.49** | **28.65** |
| | QED (*OOD*) | 59.50 | 0.48 | 11.92 |
| Trained on JNK3 | JNK3 (*In-Domain*) | **81.00** | **0.47** | **10.06** |
| | QED (*OOD*) | 76.0 | 0.51 | 22.19 |
| | DRD2 (*OOD*) | 0.00 | 0.58 | 1.16 |
| Trained on DRD2 | DRD2 (*In-Domain*) | **97.00** | **0.49** | **16.70** |
| | JNK3 (*OOD*) | 54.50 | 0.52 | 13.36 |
| | plogP (*OOD*) | 5.00 | 0.70 | 3.30 |

In-domain performance is highlighted in **bold**. The underlined value highlights the most significant case of positive knowledge transfer.

points below the QED-specialist model. This strong transfer can be explained by the complementary nature of these optimization objectives: JNK3 inhibition requires optimizing for bioactivity, which often involves improving molecular properties like appropriate lipophilicity, reasonable molecular weight, and balanced polarity — features that directly contribute to QED scores. Essentially, molecules optimized for kinase inhibition frequently satisfy drug-likeness criteria as a beneficial side effect.

In contrast, transfers between chemically orthogonal properties show minimal benefit. The DRD2 →plogP transfer (5.0% success) fails because DRD2 optimization focuses on specific receptor-ligand interactions through precise pharmacophore features, while plogP is a bulk physicochemical property determined by overall hydrophobicity. Similarly, JNK3 →DRD2 transfer completely fails (0.0% success) as these targets belong to different protein families (kinase vs. GPCR) with distinct binding site architectures and pharmacophore requirements.

The asymmetric transfer patterns further illuminate the hierarchical nature of molecular properties. The plogP →QED transfer (59.5%) substantially outperforms the reverse QED →plogP transfer (13.5%). This asymmetry reflects that lipophilicity optimization (plogP) teaches the model about hydrophobic/hydrophilic balance — a fundamental aspect of drug-likeness. Conversely, QED optimization involves balancing multiple properties simultaneously, learning strategies that are too complex and multifaceted to transfer effectively to the simpler, unidimensional task of lipophilicity optimization.

These transfer patterns suggest that POLO learns hierarchical representations of chemical space, where strategies for optimizing fundamental physicochemical properties (like lipophilicity) provide a foundation that partially transfers to composite metrics (like QED), while complex multi-property optimizations create specialized knowledge that doesn't readily decompose back to simpler tasks.

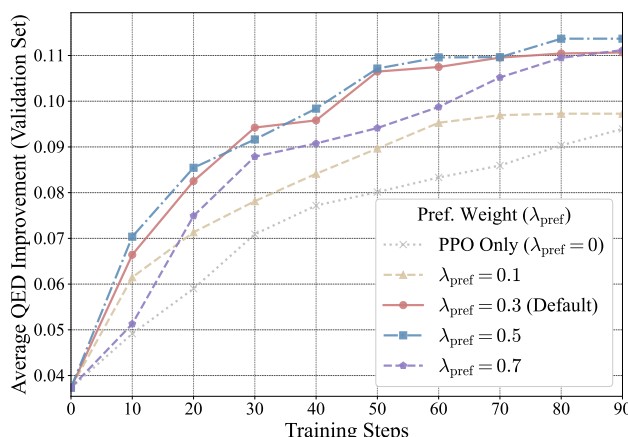

Figure 6: Analysis of preference loss weight ($\lambda_{\text{pref}}$) on validation performance.

## L  ABLATION STUDY ON PREFERENCE LOSS WEIGHT

We investigated the impact of preference loss weight $\lambda_{\text{pref}}$ on training dynamics by varying it from 0 (PPO-only) to 0.7 and evaluating on unseen validation molecules. Figure 6 shows the learning curves on the QED task.

**Preference learning is essential for generalization.** All configurations with $\lambda_{\text{pref}} > 0$ significantly outperform the PPO-only baseline on unseen data. While PPO alone plateaus at 0.09 validation improvement, adding preference learning with $\lambda_{\text{pref}} = 0.3$ reaches 0.11 — a 22% gain. This demonstrates that dense turn-level preferences provide crucial learning signals for generalizing to new molecules.

**Optimal weight balances exploration and exploitation.** The relationship between $\lambda_{\text{pref}}$ and performance is non-monotonic: performance improves from 0.1 to 0.5, then slightly decreases at 0.7. This pattern reveals an important trade-off. Moderate preference weights (0.3-0.5) optimally balance two objectives: leveraging fine-grained molecular comparisons from preferences while maintaining the exploration encouraged by PPO. Too much weight on preferences (0.7) can overshadow the trajectory-level exploration needed for discovering diverse strategies.

**Robustness across weight values.** POLO demonstrates strong robustness to the exact choice of $\lambda_{\text{pref}}$. All tested values from 0.1 to 0.7 substantially outperform the baseline, with final performance ranging from 0.097 to 0.113 — a relatively narrow band. This robustness is practically valuable as it suggests POLO will perform well without extensive hyperparameter tuning. We selected $\lambda_{\text{pref}} = 0.3$ as our default as it achieves near-optimal performance, converges quickly, and generalizes consistently across tasks.

Table 11: **Impact of Model Scale on POLO's Performance.** Performance comparison between 1.5B and 3B parameter models on `QED` task with 32 training molecules.

| Base Model | SR (%) | Sim | RI |
|---|---|---|---|
| POLO (Qwen2.5-1.5B) | 71.50 | 0.52 | 0.20 |
| POLO (Qwen2.5-3B) | **84.00** (+12.5) | **0.50** | **0.23** |

## M  IMPACT OF MODEL SCALE

To investigate the scalability of our framework, we evaluated POLO using different base model sizes on the `QED` task with only 32 training molecules.

As shown in Table 11, scaling from 1.5B to 3B parameters yields substantial improvements: success rate increases from 71.5% to 84.0% (+17.5% relative), while maintaining similar structural constraints. Notably, the larger model achieves higher success with slightly lower average similarity (0.50 vs 0.52), suggesting enhanced capability to explore beneficial yet diverse chemical modifications within the similarity constraints.

These results demonstrate two key points. First, PGPO effectively scales with model capacity, translating increased parameters into better optimization performance. Second, even with minimal training data (32 molecules), our framework can leverage larger models effectively, which is critical for practical applications where high-quality training examples are scarce.

# N    DETAILED CASE STUDIES

## N.1    OVERVIEW

We provide detailed case studies to illustrate POLO's optimization behavior across different task complexities. These examples demonstrate the complete multi-turn interaction between the agent and environment, including the agent's reasoning process (`<think>` tags) and molecular modifications (`<answer>` tags). We present both successful optimizations and challenging cases to provide a comprehensive view of the method's behavior.

## N.2    SUCCESSFUL OPTIMIZATION STRATEGIES

The successful case studies demonstrate several effective optimization patterns:

**Iterative Refinement.** In the `JNK3` optimization case, the agent learns from initial failures with simple aromatic additions and discovers that extended aromatic amine linkers significantly enhance binding affinity (0.030→0.350). This demonstrates POLO's ability to adapt strategies based on environment feedback.

**Strategic Simplification.** The `QED` optimization showcases chemical intuition where removing the indole heterocycle in favor of a simpler benzene ring improves drug-likeness (0.736→0.889). Similarly, in the triple-objective case (`DRD2 + plogP + QED`), removing a single chlorine atom simultaneously addresses multiple objectives, achieving 14.8× improvement in `DRD2` activity.

**Balanced Multi-Objective Optimization.** The `DRD2 +QED` case illustrates successful navigation of trade-offs. The agent systematically explores modifications: removing charged centers for QED improvement, then introducing p-fluorophenyl for `DRD2` binding, ultimately achieving 5× `DRD2` improvement while maintaining QED gains.

## N.3    CHALLENGING OPTIMIZATION SCENARIOS

The `DRD2 +plogP +QED` optimization case illustrates the inherent challenges of multi-objective molecular optimization with competing properties. The three objectives exhibit fundamental trade-offs: increasing `plogP` typically requires adding hydrophobic moieties, while maintaining high `QED` favors balanced polarity and moderate molecular weight. Subsequently, attempts to recover through more drastic structural changes violated similarity constraints (similarity dropping to 0.118), triggering negative rewards and preventing further exploration. This challenging case highlights that multi-objective optimization remains difficult when properties have opposing chemical requirements, a known challenge in computational drug design that affects all current methods.

## Case Study: Single-Objective Optimization (QED)

**System:** You are a molecular designer.
**User:** You are an expert medicinal chemist specializing in molecular optimization. Your task is to modify the given molecule to increase QED while keeping structural changes as minimal as possible. The modified molecule should maintain a structural similarity of at least 0.4 with the original molecule.

---

**Turn 1:**
State:
Original Molecule: Cc1cc(F)ccc1CCNC(=O)Cc1c[nH]c2c(C)cccc12
Original molecule properties: QED: 0.736
You have 5 actions left.
**Assistant:** <think>The original molecule has QED of 0.736. Try simplifying the indole system to improve drug-likeness.</think>
<answer>Cc1cc(F)ccc1CCNC(=O)Cc1ccccc1</answer>
**User:** Reward: 0.767

---

**Turn 2:**
State: Step 1 of 5
Action: Proposed 'Cc1cc(F)ccc1CCNC(=O)Cc1ccccc1'
Previous Score: 0.736
Resulting Molecule: Cc1cc(F)ccc1CCNC(=O)Cc1ccccc1
Similarity to Original: 0.604 (required >= 0.400)
New Score (drug-likeness QED): 0.889 (change: +0.153)
Great job! New best score achieved! Keep refining.
Reward: 0.767
**Assistant:** <think>Excellent improvement to 0.889. Current structure already optimized. Maintain this molecule.</think>
<answer>Cc1cc(F)ccc1CCNC(=O)Cc1ccccc1</answer>
**User:** Reward: -0.3

---

**Turn 3:**
State: Step 2 of 5
Current molecule remains: Cc1cc(F)ccc1CCNC(=O)Cc1ccccc1
Current score: 0.889
No modification detected. Your proposed SMILES matches the current molecule.
Please propose a different modification.
Reward: -0.300
**Assistant:** <think>Need a different modification. Try adding nitrile to further tune properties.</think>
<answer>Cc1cc(F)ccc1CCNC(=O)Cc1ccccc1C#N</answer>

---

**Result:** QED improved from 0.736 to $> 0.9$ (exceeding the success threshold) in just two modifications. The agent first identified that removing the indole heterocycle in favor of a simpler benzene ring significantly improved drug-likeness ($0.736 \rightarrow 0.889$). The subsequent addition of a cyano group further optimized the QED score beyond 0.9, triggering automatic task completion. This demonstrates POLO's ability to achieve rapid optimization through strategic structural simplification followed by targeted functional group additions.

## Case Study: Single-Objective Optimization (JNK3)

**System:** You are a molecular designer.
**User:** You are an expert medicinal chemist specializing in molecular optimization. Your task is to modify the given molecule to increase JNK3 inhibition while keeping structural changes as minimal as possible. The modified molecule should maintain a structural similarity of at least 0.4 with the original molecule.

---

**Turn 1:**
State: Original Molecule: Cc1c(Cl)cccc1S(=O)(=O)NCC@HCN1CCNHCC1
Original molecule properties: JNK3 inhibition: 0.030
You have 5 actions left.
**Assistant:** <think>Need to analyze the starting structure first.</think>
<answer>Cc1c(Cl)cccc1S(=O)(=O)NCC@HCN1CCNHCC1</answer>
**User:** Reward: -0.3

---

**Turn 2:**
State: Step 1 of 5
Current molecule remains: Cc1c(Cl)cccc1S(=O)(=O)NCC@HCN1CCNHCC1
Current score: 0.030
No modification detected. Your proposed SMILES matches the current molecule.
Please propose a different modification.
Reward: -0.300
**Assistant:** <think>Try adding aromatic group to piperazine.</think>
<answer>Cc1c(Cl)cccc1S(=O)(=O)NCC@HCN1CCN(c2ccccn2)CC1</answer>
**User:** Reward: -0.03

---

**Turn 3:**
State: Step 2 of 5
Action: Proposed 'Cc1c(Cl)cccc1S(=O)(=O)NCC@HCN1CCN(c2ccccn2)CC1'
Previous Score: 0.030
Similarity to Original: 0.650 (required >= 0.400)
New Score (JNK3 inhibition): 0.000 (change: -0.030)
Score decreased. Valid modification, but negatively impacted property.
Consider alternative strategies.
Reward: -0.030
**Assistant:** <think>Need larger aromatic system for JNK3 binding.</think>
<answer>Cc1c(Cl)cccc1S(=O)(=O)NCC@HCN1CCN(c2cccc(Nc3ccccn3)c2)CC1</answer>
**User:** Reward: 1.1

---

**Turn 4:**
State: Step 3 of 5
Action: Proposed 'Cc1c(Cl)cccc1S(=O)(=O)NCC@HCN1CCN(c2cccc(Nc3ccccn3)c2)CC1'
Previous Score: 0.030
Similarity to Original: 0.557 (required >= 0.400)
New Score (JNK3 inhibition): 0.250 (change: +0.220)
Great job! New best score achieved! Keep refining.
Reward: 1.100
**Assistant:** <think>Extend aromatic system further with another linker.</think>
<answer>Cc1c(Cl)cccc1S(=O)(=O)NCC@HCN1CCN(c2cccc(Nc3ccc(Nc4ccccn4)cc3)c2)CC1</answer>
**User:** Reward: 0.5

---

**Turn 5:**
State: Step 4 of 5
Action: Proposed 'Cc1c(Cl)cccc1S(=O)(=O)NCC@HCN1CCN(c2cccc(Nc3ccc(Nc4ccccn4)cc3)c2)CC1'
Previous Score: 0.250
Similarity to Original: 0.534 (required >= 0.400)
New Score (JNK3 inhibition): 0.350 (change: +0.100)
Great job! New best score achieved! Keep refining.
Reward: 0.500
You have 1 actions left.
**Assistant:** <think>Keep improving the inhibition.</think>
<answer>Cc1c(Cl)cccc1S(=O)(=O)NCC@HCN1CCN(c2cccc(Nc3ccc(Nc4ccccn4)cc3)c2)CC1</answer>

---

**Result:** JNK3 inhibition improved from 0.030 to 0.350 ($11.7\times$ improvement) while maintaining similarity of 0.534. The agent learned from early failures and discovered that aromatic amine linkers significantly enhance target binding.

---

### Case Study: Multi-Objective Optimization (DRD2 + QED)

**System:** You are a molecular designer.
**User:** You are an expert medicinal chemist specializing in molecular optimization. Your task is to modify the given molecule to increase DRD2 inhibition and increase QED while keeping structural changes as minimal as possible. The modified molecule should maintain a structural similarity of at least 0.4 with the original molecule.

---

**Turn 1:**
State:
Original Molecule: C[C@H]1CCC[C@@H](NC(=O)CN2CC[NH+](Cc3cccc4cn[nH]c34)CC2)C1
Original properties: DRD2: 0.099, QED: 0.735
You have 5 actions left.
**Assistant:** <think>Replace charged NH+ and indazole with neutral aromatic.</think>
<answer>C[C@H]1CCC[C@@H](NC(=O)CN2CCN(Cc3ccc(C)cc3)CC2)C1</answer>
**User:** Reward: 1.279

---

**Turn 2:**
State: Step 1 of 5
Similarity: 0.500, DRD2: 0.023 ($\downarrow$0.076), QED: 0.893 ($\uparrow$0.158)
Great QED improvement! Threshold achieved (+0.158 $\geq$ 0.1)!
However, DRD2 activity decreased. Try to improve DRD2 inhibition.
**Assistant:** <think>Replace p-tolyl with p-fluorophenyl for better DRD2 binding.</think>
<answer>C[C@H]1CCC[C@@H](NC(=O)CN2CCN(c3ccc(F)cc3)CC2)C1</answer>
**User:** Reward: 2.041

---

**Turn 3:**
State: Step 2 of 5
Similarity: 0.486, DRD2: 0.146 ($\uparrow$0.047), QED: 0.920 ($\uparrow$0.186)
Great DRD2 activity improvement! Great QED improvement!
Excellent progress! Keep refining with this approach.
**Assistant:** <think>Add extra carbon to cyclohexyl ring for better fit.</think>
<answer>C[C@H]1CCC[C@@H](NC(=O)CN2CCN(c3ccc(F)cc3)CC2)CC1</answer>
**User:** Reward: 2.631

---

**Turn 4:**
State: Step 3 of 5
Similarity: 0.405, DRD2: 0.466 ($uparrow$0.367), QED: 0.851 ($uparrow$0.116)
Great DRD2 activity improvement! Excellent progress!
**Assistant:** <think>Current molecule performing well, maintain.</think>
<answer>C[C@H]1CCC[C@@H](NC(=O)CN2CCN(c3ccc(F)cc3)CC2)CC1</answer>
**User:** Reward: -0.3

---

**Turn 5:**
No modification detected. Please propose a different modification.
**Assistant:** <think>Try adding phenyl extension for additional interactions.</think>
<answer>C[C@H]1CCC[C@@H](NC(=O)CN2CCN(c3ccc(F)cc3)CC2)CC1=Cc1ccccc1</answer>

---

**Result:** Successfully balanced dual objectives: DRD2 improved from 0.099 to 0.466 (5$\times$ improvement) and QED from 0.735 to >0.85 (+16%). Key modifications: (1) removing the charged piperazine improved QED, (2) p-fluorophenyl direct attachment enhanced DRD2 binding, (3) ring expansion to cycloheptyl optimized receptor fit. This demonstrates POLO's ability to navigate trade-offs in multi-objective optimization.

---

### Case Study: Triple-Objective Optimization (DRD2 + `plogP` + `QED`)

**System:** You are a molecular designer.
**User:** You are an expert medicinal chemist specializing in molecular optimization. Your task is to modify the given molecule to increase DRD2 inhibition, increase LogP and increase QED while keeping structural changes as minimal as possible. The modified molecule should maintain a structural similarity of at least 0.4 with the original molecule.

---

**Turn 1:**
State:
Original Molecule: COc1ccc(O)c(CNC2CC[NH+](Cc3ccccc3Cl)CC2)c1
Original properties: DRD2: 0.061, LogP: -0.095, QED: 0.740
You have 5 actions left.
**Assistant:** <think>Need to increase LogP and QED. Try major structural
reorganization with cyclopropyl and longer alkyl chain.</think>
<answer>CCCN(Cc1ccc(CC2CC2)cc1)C1CCN(Cc2ccccc2Cl)CC1</answer>
**User:** Reward: -0.326

---

**Turn 2:**
State: Step 1 of 5
Similarity too low: 0.237 < required 0.400
Current molecule remains: COc1ccc(O)c(CNC2CC[NH+](Cc3ccccc3Cl)CC2)c1
Consider smaller, more conservative changes.
**Assistant:** <think>Too aggressive. Simply remove Cl to reduce polarity.</think>
<answer>COc1ccc(O)c(CNC2CCN(Cc3ccccc3)CC2)c1</answer>
**User:** Reward: 17.674

---

**Turn 3:**
State: Step 2 of 5
Similarity: 0.586, DRD2: 0.901 ($uparrow$0.840), LogP: 2.063 ($uparrow$2.157), QED: 0.856 ($uparrow$0.116)
OUTSTANDING! All targets achieved! Success bonus: +11.0
DRD2 threshold achieved (+0.840 $\geq$ 0.5)!
LogP threshold achieved (+2.157 $\geq$ 1.0)!
QED threshold achieved (+0.116 $\geq$ 0.1)!
**Assistant:** <think>Perfect! All objectives met. Maintain this structure.</think>
<answer>COc1ccc(O)c(CNC2CCN(Cc3ccccc3)CC2)c1</answer>
**User:** Reward: -0.3

---

**Result:** Achieved remarkable simultaneous improvement across all three objectives with a single atom deletion: DRD2: 0.061→0.901 (14.8× improvement), LogP: -0.095→2.063 (+2.16 units), QED: 0.740→0.856 (+16%). The key insight was that removing the ortho-chlorine eliminated both the charged piperazine (improving QED and LogP) and steric clash (dramatically improving DRD2 binding). This demonstrates how a single, well-chosen modification can address multiple objectives when the problematic feature affects all targets.

---

### Case Study: Failed Multi-Objective Optimization (DRD2 + `plogP` + QED)

**System:** You are a molecular designer.
**User:** Your task is to modify the given molecule to increase DRD2 inhibition, increase LogP and increase QED while keeping structural changes as minimal as possible.

---

**Turn 1:**
Original: CN1CCO[C@@H]([C@H](O)Cc2ccncc2)C1
Properties: DRD2: 0.035, LogP: -1.631, QED: 0.800
**Assistant:** <think>LogP is quite low. Add more bulky groups.</think>
<answer>CCCCN1CCO[C@@H]([C@H](O)Cc2ccncc2)CC1</answer>
**User:** Reward: -0.668

---

**Turn 2:**
DRD2: 0.240 (↑0.205), LogP: -3.750 (↓2.119), QED: 0.863 (↑0.063)
LogP decreased. Try to improve LogP.
**Assistant:** <think>Add conjugated system for LogP.</think>
<answer>CCCc1ccccc1C</answer> % Complete structure change - loses similarity
**User:** Reward: -0.604

---

**Turn 3:**
Environment reverted to best molecule (step 1)
**Assistant:** <think>Try longer conjugated system.</think>
<answer>CCCc1ccc(-c2ccccc2)cc1</answer>
**User:** Reward: -0.565

---

**Turn 4:**
Similarity too low: 0.118 < required 0.400
**Assistant:** <think>Keep minimal changes, focus on LogP.</think>
<answer>CCCCN1CCO[C@@H]([C@H](O)Cc2ccc(F)cc2)CC1</answer> % Changes pyridine to fluorobenzene
**User:** Reward: 0.809

---

**Turn 5:**
Carbon chain too long: 11 atoms (limit ≤ 10)
**Assistant:** <think>Make more conservative changes.</think>
<answer>CCCNC1CCO[C@@H]([C@H](O)Cc2ccc(F)cc2)CC1</answer>

---

**Result:** Failed to achieve balanced optimization. The agent struggled with conflicting objectives: attempts to improve LogP (adding hydrophobic groups) decreased it further due to unintended polarity changes. Multiple similarity violations and structural constraint violations occurred. This case illustrates challenges in multi-objective optimization when properties have opposing chemical requirements, and highlights the importance of understanding structure-property relationships beyond simple heuristics.

O  VISUALIZATION OF CASE STUDIES

Table 12: **Examples of molecular optimization trajectories.**

| Task | Original | | Edited | | Final | |
|------|----------|----------|--------|----------|-------|----------|
| | **Molecule** | **Properties** | **Molecule** | **Properties** | **Molecule** | **Properties** |
| QED |  | QED: 0.77 |  | QED: 0.86
Sim: 0.54 |  | QED: 0.94
Sim: 0.51 |
| plogP |  | plogP: -0.09 |  | plogP: 2.10
Sim: 0.51 |  | plogP: 2.44
Sim: 0.56 |
| DRD2 |  | DRD2: 0.05 |  | DRD2: 0.96
Sim: 0.79 |  | DRD2: 0.96
Sim: 0.79 |
| QED + SA |  | QED: 0.78
SA: 3.59 |  | QED: 0.87
SA: 2.86
Sim: 0.54 |  | QED: 0.91
SA: 2.82
Sim: 0.50 |
| QED + DRD2 |  | QED: 0.76
DRD2: 0.06 |  | QED: 0.85
DRD2: 0.03
Sim: 0.66 |  | QED: 0.76
DRD2: 0.65
Sim: 0.45 |
| SA + DRD2 |  | SA: 3.57
DRD2: 0.16 |  | SA: 2.53
DRD2: 0.50
Sim: 0.51 |  | SA: 2.42
DRD2: 0.88
Sim: 0.58 |

