# OpenReview forum: "POLO: Preference-Guided Multi-Turn Reinforcement Learning for Lead Optimization"
_ICLR.cc/2026/Conference — Submitted to ICLR 2026_

### Official Review · Reviewer_xJaw · 2025-10-19

**Soundness:** 3
**Presentation:** 2
**Contribution:** 2
**Rating:** 4
**Confidence:** 4

**Summary:**

The paper introduces a framework that adapts large language models (LLMs) for sample-efficient molecular lead optimization. Unlike prior approaches that optimize molecules one step at a time, POLO models the process as a multi-turn reinforcement learning (RL) problem, where the agent iteratively proposes modifications to lead compounds. Its core algorithm, Preference-Guided Policy Optimization (PGPO), combines trajectory-level policy optimization with turn-level preference learning, enabling the extraction of O(NT²) training signals from N trajectories of length T, rather than just O(N) scalar rewards. The method leverages oracle feedback for properties such as QED, plogP, and bioactivity, while enforcing similarity constraints to maintain drug-likeness. Experiments on ten optimization tasks show that POLO significantly outperforms baselines—including GA, REINVENT, and chemistry-specific LLMs—achieving up to much higher success rates within a budget of 500 oracle evaluations. Overall, the paper demonstrates that combining multi-turn RL with preference learning yields state-of-the-art sample efficiency and robustness for molecular optimization.

**Strengths:**

1. **Essential problem domain**: The paper tackles lead optimization in drug discovery, a field of high scientific and practical importance where each oracle evaluation can be very costly.
2. **Clear motivation and logical presentation**: The authors present a well-structured narrative: they identify the limitations of existing RL and LLM-based methods, motivate the need for multi-turn reasoning, and clearly explain how Preference-Guided Policy Optimization (PGPO) fills this gap. The logic connecting problem setup, method design, and evaluation is easy to follow.
3. **Interesting method with validation**: The dual-level learning approach—combining trajectory-level PPO with turn-level preference learning—is novel and interesting. The method is validated across multiple tasks. While there are some issues (in Weaknesses 2), the experiments provide convincing support for the claimed benefits.

**Weaknesses:**

1. **Necessity of the RL and preference design**: In this setup, every molecule can be directly assigned a scalar score by the oracle. A straightforward design would be to use these scores as dense signals for training. The introduction of preference-based learning raises the question of whether relative comparisons are strictly necessary here, as they may discard some information from the original scores.
2. **Evaluation setup**:
   - **Oracle budget**: The budget of 500 evaluations is arguably too low compared with realistic practice. For example, the PMO benchmark [1] adopts a budget of 10,000 evaluations. The strict budget in this paper may disproportionately disadvantage strong non-LLM baselines such as Reinvent 4.
   - **Task complexity**: The chosen oracles (QED, plogP, SA, DRD2, JNK3) have been criticized as relatively simple, and pretrained LLMs may already encode prior knowledge about them. More challenging objectives, such as docking scores [2], would provide a more realistic and convincing testbed.
3. **Variability across leads**: Different initial lead molecules can yield highly variable optimization outcomes. Reporting only average performance risks obscuring this variability. Including statistics such as standard deviations or distribution plots would give a clearer picture of robustness across leads.
4. **Minor issues**:
   - Garbled characters appear in Table 1.
   - Figures 3 and 4 seem to be mislabeled.
   - A discussion of the limitations of the work would make the paper more balanced.

[1] Sample Efficiency Matters: A Benchmark for Practical Molecular Optimization.

[2] Generative Models Should at Least Be Able to Design Molecules That Dock Well: A New Benchmark.

**Questions:**

1. **SMILES randomization**: How does the method handle randomized (non-canonical) SMILES representations, which are widely used as a form of data augmentation in molecular modeling? Would randomization affect stability or performance during optimization?
2. **Dependence on prior knowledge**: To what extent does POLO’s performance depend on the prior chemical knowledge already encoded in the pretrained LLM?
3. **Scaling with number of leads**: In Figure 4 (d), the improvement from increasing the number of lead molecules appears not to have plateaued. Why did the experiments stop at this scale, and what do the authors expect would happen if the number of leads were further increased?

---

> ### Author Response · Authors · 2025-11-23
> **Response to Reviewer xJaw Part 1/2**
>
> We thank the reviewer for their constructive feedback. We are encouraged that the reviewer recognizes the importance of the problem domain and finds our dual-level learning approach interesting. We address the specific concerns regarding method necessity, evaluation setup, and variability below with detailed clarifications and additional data.
>
>  ## W1: Necessity of Preference Design
>
> We appreciate the reviewer's question regarding the necessity of preference learning. We clarify that our framework utilizes **both absolute scalar scores and relative preference signals**, as they provide complementary guidance.
>
> *   **Dual-Level Mechanism:** As detailed in **Section 2.4**, our dual-level design leverages scalar scores directly in the trajectory-level PPO component ($J_{traj}$) to reinforce successful long-term strategies. However, relying solely on scalar rewards extracts only $O(N)$ signals from $N$ trajectories. By augmenting this with turn-level preference learning ($J_{pref}$), we extract fine-grained, comparative feedback from intermediate steps (e.g., ''modification A is better than B'').
> *   **Signal Amplification:** Relying solely on scalar rewards yields only $O(N)$ signals. By incorporating pairwise preferences, we achieve a quadratic amplification ($O(NT^2)$), maximizing the information extracted from each costly oracle call (see **Appendix I**).
> *   **Empirical Validation:** Our ablation study (**Table 3**) empirically validates this design: removing the preference component (and relying only on dense scalar rewards) results in a significant performance drop (e.g., QED Success Rate falls from **91.0% to 81.5%**). This confirms that relative preference signals provide distinct, additive value beyond scalar rewards alone.
>
>  ## W2: Evaluation Setup
>
> **1. Oracle Budget:**
> We appreciate the reviewer's perspective on the oracle budget. We would like to clarify that the strict budget of 500 calls is a deliberate design choice aligned with the central theme of our paper: **Sample Efficiency**.
>
> *   **Wet-Lab Reality:** While benchmarks like PMO allow 10,000 calls, such volume is prohibitive in real-world wet-lab optimization, where each synthesis and test incurs significant cost.
> *   **Addressing the Bottleneck:** Our goal is specifically to address this bottleneck. The ability of POLO to identify high-quality candidates within just 500 evaluations demonstrates precisely the sample-efficient capability required to accelerate drug discovery.
>
> **2. Task Complexity:**
> We respectfully clarify that our chosen tasks, particularly JNK3 and DRD2, represent significant chemical challenges.
> *   **Bioactivity Challenge:** JNK3 and DRD2 tasks require optimizing **protein-ligand binding affinity**. This involves learning complex Structure-Activity Relationships (SAR) to fit specific receptor pockets, which is fundamentally harder than optimizing heuristic properties like QED.
> *   **Validation via Cited Benchmark:** The difficulty of these targets is confirmed by the benchmark cited by the reviewer ([2] Cieplinski et al.).
>     *   Their work explicitly highlights that optimizing for binding (docking) is a formidable challenge where many SOTA models fail.
>     *   Crucially, they include **D2 (DRD2)** as a primary test case.
>     Our method achieves **97% Success Rate** on this exact target (DRD2) under strict similarity constraints, effectively solving a problem identified by the community as difficult.
> *   **Comprehensive Scope:** Beyond these difficult bioactivity tasks, our evaluation spans 10 diverse scenarios, including multi-objective optimization, ensuring a rigorous assessment across the full spectrum of lead optimization complexity.
>
>  ## W3: Variability Across Leads
>
> We agree that analyzing variability is important given the diverse starting points. To address this, we analyzed the distribution of optimization outcomes across the 200 lead molecules for two tasks.
>
> | Task | Initial Value | Final Value | Final Value CV | Absolute Improvement |
> | :--- | :---: | :---: | :---: | :---: |
> | **DRD2** | 0.068 ± 0.040 | 0.944 ± 0.071 | 7.47% | 0.876 ± 0.082 |
> | **QED** | 0.754 ± 0.029 | 0.925 ± 0.018 | 1.90% | 0.171 ± 0.031 |
>
> *   **Consistent Optimization:** Despite variability in initial values, the Final Value shows remarkably low variance. The Coefficient of Variation ($CV = \sigma / \mu$) is just **7.47%** for DRD2 and **1.90%** for QED. This low relative variability confirms that POLO consistently optimizes diverse leads to a uniformly high-quality state, rather than depending on specific "easy" starting points. We will include these detailed statistics and distribution plots in the Appendix.
>
>
>  ## W4: Minor Issues
>
> We thank the reviewer for spotting these errors. We have corrected the character encoding issues in Table 1 and fixed the labeling for Figures 3 and 4. We will expand the limitations section to discuss constraints.

---

> ### Author Response · Authors · 2025-11-23
> **Response to Reviewer xJaw Part 2/2**
>
> ## Q1: SMILES Randomization
>
> We clarify that our framework strictly enforces **Canonicalization** at both training and inference stages. Before any processing (input to LLM) or evaluation (oracle scoring), all SMILES strings are converted to their canonical form using RDKit. This standardization ensures a one-to-one mapping between molecular structure and string representation, eliminating ambiguity and ensuring stability during the optimization process.
>
>  ## Q2: Dependence on Prior Knowledge
>
> We acknowledge that prior knowledge (via SFT) is a critical foundation for sample efficiency, though its impact varies by task difficulty.
>
> **1. Necessity and Task Dependence:**
> SFT provides the ''cold start'' capability essential for our tight budget.
> *   **Simple Tasks (e.g., QED):** The dependence is lower. Even without SFT initialization, the model can achieve reasonable performance via RL exploration alone (71.0% SR, Table 3).
> *   **Complex Tasks (e.g., DRD2):** The dependence is higher. SFT provides crucial priors for bioactivity, boosting success rates significantly (41.5% $\to$ 97.0%).
>
>
> **2. RL Unlocks the Prior:**
> Prior knowledge alone is insufficient. Baselines relying solely on priors (e.g., `SFT Only`, `GeLLM3O`) plateau at ~63% on DRD2. It is our **PGPO algorithm** that effectively exploits this prior to reach **97%** success.
>
> **3. Generalization Context:**
> We emphasize that our SFT is a **one-time offline process** on a dataset strictly separated from our test tasks to prevent leakage. Unlike baselines that optimize *on the test instance* (e.g., **Reinvent 4, Graph-GA**), POLO must generalize to unseen molecules. SFT provides the necessary chemical representation for this more challenging inference setting.
>
>
>  ## Q3: Scaling with Number of Leads
>
> We appreciate the reviewer's concern about the scaling behavior of our model. To fully address this, we conducted additional experiments with larger training sets (up to 512 leads), as summarized below:
>
> | Training Size | QED SR (%) | plogP SR (%) | DRD2 SR (%) |
> | :---: | :---: | :---: | :---: |
> | **32** | 71.5 | 68.0 | 78.0 |
> | **64** | 81.5 | 85.0 | 85.0 |
> | **128** | 92.0 | 99.0 | 97.0 |
> | **256** | 94.5 | 100.0 | 98.0 |
> | **512** | 96.0 | 100.0 | 98.0 |
>
> *   **Performance Trends:** As shown above, performance does improve with more data but eventually saturates (e.g., DRD2 plateaus at ~98% after 256 leads). Since 128 leads already yield near-optimal results, we focused on this efficient regime, which is most relevant for practical scenarios where high-quality lead data is scarce.
> *   **Core Insight (Data Efficiency):** We wish to highlight that the primary motivation for Figure 4(d) is to demonstrate **Data Efficiency**. Even with minimal data (only 32 leads), POLO achieves strong results (71.5% QED). This validates our claim that PGPO maximizes signal extraction ($O(NT^2)$) from limited data.
>
> We will update the figure in the revision to include these extended results.

---

### Official Review · Reviewer_LZd8 · 2025-10-27

**Soundness:** 2
**Presentation:** 3
**Contribution:** 3
**Rating:** 4
**Confidence:** 3

**Summary:**

This submission focuses on enhancing the ability of large language models (LLMs) on the molecular optimization task. Notably, prior approaches focus on enhancing the model's performance via in-context learning, instead of leveraging the iterative nature of the molecular optimization task. These paradigms lead the LLM to learn from isolated optimization steps instead of the complete trajectories, thereby failing to fully utilize the past experiences. To address this, this submission proposes preference-guided multi-turn optimization for lead optimization (POLO), which introduces preference-guided policy optimization to extract learning signals from trajectory-level (traj-level) and turn-level to fully utilize the reasoning trajectory to enhance the performance. The submission conducts extensive empirical studies and justifies the effectiveness of the proposed method.

**Strengths:**

- The motivation of this submission is clear: by combining turn-level and traj-level loss, the model can learn better from past experience to optimize the molecule better.
- The submission is generally well-written, with clear illustrations and tables.
- Extensive experiments have been conducted to provide a good insight into the components of the proposed method.

**Weaknesses:**

- The proposed turn-level and trajectory-level losses are interesting but require further justification. The interpolation between on-policy RL (PPO) and off-policy RL (DPO) may introduce gradient bias, which is not theoretically analyzed or empirically verified.
- The description of the turn-level loss is unclear. The optimization target appears to be defined on the state incorporating previous history, which could lead to double-counting the contribution of earlier states to the final outcome. Moreover, the LLM may not be able to identify which specific actions are responsible for the observed reward improvement, raising concerns about proper credit assignment.
- The performance comparisons appear unfair. POLO is trained directly on the target properties using the constructed MolOptIns dataset, whereas other LLM-based methods rely only on prompt-based adaptation. Additionally, POLO employs evolutionary search during inference, which provides an extra advantage. It would be more convincing to evaluate other task-specific LLMs under the same inference setup to fairly assess the effectiveness of the proposed turn-level loss. Furthermore, according to the original MOLLO paper, it achieves relatively strong performance, yet in this submission, the success rate on JNK3 is reported as zero, which raises questions about the evaluation setup or implementation details.

**Questions:**

1. What guarantees or empirical evidence show that the mixed objective preserves a valid policy gradient direction?
2. How does the algorithm prevent redundant gradient accumulation from overlapping state representations?
3. Discuss the fairness of the experiment setting, or provide additional experiments to justify to effectiveness on POLO.

---

> ### Author Response · Authors · 2025-11-23
> **Response to Reviewer LZd8 Part 1/3**
>
> We thank the reviewer for the constructive feedback. We appreciate the recognition of our motivation and the extensiveness of our experiments. We address the specific concerns raised in the review below, providing detailed clarifications and additional evidence to strengthen our claims.
>
>  ## W1 & Q1: Theoretical Validity and Gradient Bias
>
> We appreciate the reviewer's insightful question regarding gradient validity. We would like to clarify the on-policy nature of our implementation and provide additional empirical evidence to demonstrate the stability of the mixed objective.
>
> **1. Clarification: Fully On-Policy Learning**
>
> The concern regarding bias arises from the assumption of mixing on-policy PPO with off-policy DPO. We clarify that our preference learning is also **strictly on-policy**:
> *   Unlike offline DPO (fixed dataset), we construct preference pairs dynamically from the **current policy's trajectories** ($\tau \sim \pi_\theta$) at each iteration.
> *   Since both objectives operate on the exact same on-policy distribution, there is **no distribution mismatch or off-policy bias**. The two losses are linearly combined ($\nabla L = \nabla L_{PPO} + \lambda \nabla L_{pref}$), preserving a valid update direction.
>
> **2. Empirical Evidence: Synergy and Alignment**
> *   **Performance-Level (Existing):** As shown in our ablation study (**Table 3**), removing either component degrades performance significantly, confirming that the mixed objective improves rather than destabilizes the policy.
> *   **Gradient-Level (New Analysis):** To directly verify gradient validity, we analyzed the update vectors during training. We found a **cosine similarity of 0.954** between $\nabla L_{PPO}$ and $\nabla L_{pref}$, confirming that the two signals are highly aligned and cooperatively guide the policy toward higher-reward regions.

---

> ### Author Response · Authors · 2025-11-23
> **Response to Reviewer LZd8 Part 2/3**
>
> ## W2 & Q2: Double Counting and Credit Assignment
>
> We thank the reviewer for the detailed feedback on the loss formulation. We clarify that our implementation inherently prevents double-counting.
>
> *   **Stepwise Independence:** As defined in **Section 2.4.2**, the preference term relies on the implicit reward $\psi_t = \beta \log \frac{\pi_\theta(a_t|s_t)}{\pi_{ref}(a_t|s_t)}$. Crucially, we employ a **stepwise** calculation mode where the score for a turn $t$ is the sum of implicit rewards **only for the tokens within that specific turn span**. This strictly differs from a cumulative mode that aggregates history. Because each turn's score is computed independently, the gradient for a preference pair $(t_i, t_j)$ updates only the policy distributions for actions in those specific turns, without re-accumulating gradients from earlier history states.
>
> *   **Precise Credit Assignment:** This stepwise independence directly enables precise credit assignment. Since the loss function specifically reinforces the actions in turn $j$ that led to the improved Oracle Reward (e.g., QED score), credit is attributed to the specific modification rather than the entire history.
>
> We will include these implementation details in the Appendix to further clarify the mechanism.

---

> ### Author Response · Authors · 2025-11-23
> **Response to Reviewer LZd8 Part 3/3**
>
> ## W3 & Q3: Fairness of Comparison
>
> We appreciate the reviewer's careful examination of our experimental setup. We address the concerns regarding fairness in training, inference strategies, and the specific performance of MOLLEO below.
>
> **1. Training Fairness & Generalization**
>
> The reviewer raised a concern that comparing POLO (SFT-trained) against prompt-based LLMs might be unfair. We clarify that:
> *   **SFT Baselines Are Included:** Our baseline set includes **multiple SFT or Task-Specific LLMs**, such as **LlaSMol**, **ChemLLM**, **PEIT-LLM**, and **GeLLM3O**. These models were also fine-tuned on chemical instructions/data, ensuring a fair comparison of domain-adapted models.
> *   **POLO Faces a Harder Generalization Setting:** POLO is trained once as a generalist policy and evaluated on unseen test molecules. In contrast, traditional baselines like **Graph-GA**, **MOLLEO**, and **Reinvent 4** are instance-specific or online methods, restarting optimization **from scratch** for each new molecule. Despite this harder setting, POLO still achieves superior efficiency.
>
> **2. Inference Fairness (Evolutionary Search)**
>
> The reviewer expressed concerns about the fairness of using evolutionary inference. We would like to emphasize the following:
>
> * **Evolutionary Search Is Consistent Across Methods**: Baselines such as **Graph-GA** and **MOLLEO** are themselves evolutionary algorithms. Using evolutionary inference does not provide POLO with an unfair advantage; rather, it enables a structurally comparable evaluation.
>
> * **Core Algorithmic Gains Are Independent of Evolutionary Search**: Even without evolutionary inference, our core algorithm **PGPO** outperforms all baselines. As shown in the table below (excerpted from **Table 1** and **Table 3**), **POLO (w/o Evolutionary Inference)** achieves significantly higher Success Rates (SR) than the best baselines:
>
> | Model | QED SR (%) | DRD2 SR (%) |
> | :--- | :---: | :---: |
> | **Best non-LLM Baseline (Graph-GA)** | 59.50 | 34.00 |
> | **Best LLM Baseline (GeLLM3O)** | 61.50 | 49.00 |
> | **POLO (w/o Evolutionary Inference)** | **89.00** | **82.50** |
> | **POLO (Full)** | **91.00** | **97.00** |
>
> This confirms that our performance gain is primarily driven by the **PGPO algorithm**, not the inference-time search.
>
>
> **3. Discrepancy in MOLLEO Results**
>
> We appreciate the reviewer noting this discrepancy. We clarify that the 0% success rate arises from fundamental differences in task definition and evaluation metrics between the two studies:
> *   **Our Setting (Sample-Efficient Lead Optimization):** We focus on a realistic, difficult setting requiring **Success Rate (SR)** under strict constraints: a limited budget (500 calls) and high structural similarity ($\ge 0.4$) to the lead.
> *   **MOLLEO's Setting (Unconstrained Global Optimization):** MOLLEO uses a **Top-10 AUC** metric with a generous budget (10,000 calls) and, crucially, **no similarity constraint** for the JNK3 task. This encourages finding high-scoring molecules regardless of how much they deviate from the starting point.
>
> **Why MOLLEO Fails Here:**
> Since MOLLEO prioritizes score maximization without similarity constraints, its generated molecules drift significantly from the lead. As reported in the **MOLLEO paper (Appendix D.5, Table A13)**, the average similarity of edited molecules is extremely low (e.g., **0.173** for BioT5). Consequently, under our rigorous criterion (Similarity $\ge 0.4$), these candidates are classified as failures, resulting in a 0% Success Rate.
>
> This indicates that while MOLLEO excels at unconstrained molecular generation, it is not well-suited to similarity-constrained lead optimization, which is the core of our benchmark.

---

> > ### Comment · Reviewer_LZd8 · 2025-11-27
> >
> > I thank the author for the responses. Most of my concerns are addressed.
> >
> > For W3&Q3, my concern is that, for models like LlaSMol and GeLLM3O, they are not trained on optimization targets like JNK3 and SA, while POLO is trained on the task-specific dataset MolOptIns (Appendix E). In addition, the authors mention that "Chemical priors enable effective learning" in Sec. 3.5, further justifying the necessity of training on the task-specific datasets. Therefore, I suggest the authors include the model trained on MolOptIns using the same backbone and training configurations of POLO for a fair comparison.

---

> > > ### Author Response · Authors · 2025-12-03
> > > **Response to Reviewer LZd8's Follow-up**
> > >
> > > We thank the reviewer for this question. We clarify that the requested
> > > comparison already exists in our ablation study and provide additional
> > > details below.
> > >
> > > **1. Ablation Study Already Provides This Comparison**
> > >
> > > In Table 3, we provide comprehensive ablation studies on our method's
> > > components. The "w/o multi-turn" variant is **exactly the SFT-only
> > > baseline** the reviewer requests, using the same **Qwen2.5-1.5B** backbone,
> > > the same **MolOptIns** dataset, and the same training configuration,
> > > but without PGPO.
> > >
> > > This setup is essentially **equivalent to running LlaSMol or GeLLM3O on
> > > our dataset**, as they follow the same paradigm of base LLM plus
> > > domain-specific SFT. The main results on QED and DRD2 are reported in
> > > Table 3. We supplement with JNK3 and SA results below:
> > >
> > > | Model | QED SR | plogP SR | DRD2 SR | JNK3 SR | SA SR |
> > > |-------|--------|----------|---------|---------|-------|
> > > | SFT-only (w/o PGPO) | 62.0% | 53.0% | 39.5% | 0% | 11.5% |
> > > | POLO (Full) | 91.0% | 99.0% | 97.0% | 81.0% | 53.5% |
> > >
> > > **2. SFT Is Necessary but Insufficient**
> > >
> > > SFT provides an important foundation for molecular optimization.
> > > Following the practice of GeLLM3O, we use SFT to teach the model basic
> > > capabilities, specifically generating molecules with **high structural
> > > similarity and incremental property improvements** per step.
> > >
> > > However, SFT alone has a critical limitation for challenging tasks.
> > > SFT-trained models, including GeLLM3O, tend to generate molecules that
> > > are structurally similar but **fail to reach the success threshold**.
> > > For example, on JNK3, the success threshold requires a score of at least
> > > **0.4**, but the SFT-only model achieves a maximum score of only **0.37**.
> > > The model generates many high-similarity candidates, but none cross the
> > > property threshold, resulting in **0% success rate**. This occurs because
> > > SFT only learns from single-step demonstrations and cannot discover
> > > the multi-step refinement strategies needed to achieve larger
> > > cumulative improvements.
> > >
> > > **3. PGPO Provides the Critical Gains**
> > >
> > > The substantial improvements (**+81% on JNK3, +57.5% on DRD2, +42% on SA**)
> > > come entirely from PGPO's multi-turn learning mechanism. By learning
> > > from complete optimization trajectories, PGPO enables the model to
> > > discover effective **multi-step modification strategies** and accumulate
> > > incremental gains across turns to cross success thresholds. This is
> > > precisely why existing SFT-based methods struggle on hard tasks like
> > > JNK3, as they lack the capability to learn iterative refinement from
> > > trajectory-level feedback.
> > >
> > >
> > > We will include these results and findings in the revised version.

---

### Official Review · Reviewer_ZM6i · 2025-10-31

**Soundness:** 2
**Presentation:** 3
**Contribution:** 2
**Rating:** 2
**Confidence:** 4

**Summary:**

The paper introduces POLO, a framework that enables large language models to perform sample-efficient molecular lead optimization by learning from complete optimization trajectories rather than isolated steps.
Its core algorithm, Preference-Guided Policy Optimization, integrates trajectory-level reinforcement learning, capturing long-term strategies, with turn-level preference learning, which ranks intermediate molecules to provide dense, comparative feedback.
This dual-level optimization extracts maximal learning from each costly oracle call, dramatically improving efficiency.
Built on multi-turn reinforcement learning and augmented by similarity-aware instruction tuning, trajectory filtering, and evolutionary inference, POLO claims to reach the state-of-the-art performance surpass existing LLM- and non-LLM-based baselines.

**Strengths:**

This paper introduces the Preference-Guided Policy Optimization (PGPO) algorithm, which unifies trajectory-level reinforcement learning with turn-level preference learning to extract dense supervision from inherently sparse oracle feedback.
This paper is well structured. It also presents comprehensive ablation studies and strong empirical validation across both single- and multi-property molecular optimization tasks, reinforcing the robustness of its approach. The results demonstrate remarkable performance improvements—up to 2.3× better than baselines—while preserving chemical validity and structural similarity, highlighting both reliability and reproducibility. The paper clearly illustrates the algorithm and how multi-turn reasoning enhances optimization efficiency and ultimately establishes POLO as a generalizable paradigm for applying agentic LLMs to iterative, data-constrained scientific discovery.

**Weaknesses:**

I would like to raise the following concerns about this paper,

1. *Overclaiming:* The paper claims state-of-the-art performance but does not compare against established molecular or lead optimization methods, including ML-based (e.g., Random Forest), GPT-based, and Bayesian optimization approaches. Conceptually, it is difficult to justify how an LLM-driven lead optimization framework could consistently outperform specialized models designed for multi-objective reward optimization.

2. *Limited novelty:* The integration of preference learning into multi-turn reinforcement learning has already been explored in prior works such as Shani et al. (NeurIPS 2024) [1]. The proposed approach does not clearly extend or theoretically improve upon these frameworks.

3. *Potential Deficit in Algorithm design:* The algorithm simply combines multi-turn preference learning with trajectory-level optimization, but lacks theoretical justification on how this integration benefits policy improvement. In fact, introducing both trajectory-level rewards O(N) and multi-turn preferences O(NT²) could lead to conflicting optimization directions and unnecessary computational complexity within each policy-update step, potentially weakening convergence. [2]  By hacking the \lambda value could make the result seems ok, but it's not theoretically ground.

4. *Limited theoretical justification:* The PGPO formulation lacks a rigorous analysis explaining why combining trajectory- and turn-level signals enhances sample efficiency or stability.

5. *Missing related work discussion and citations:* Although this paper focuses on LLM-based reasoning, it overlooks several closely related studies on GPT-based molecular optimization. These works are not cited or discussed, leaving the positioning of this study within the existing literature unclear.

6. *Insufficient evaluation diversity:* Experiments are restricted to small-molecule optimization tasks (QED, pLogP, DRD2 on ZINC-250k) without testing scalability to larger, more chemically diverse datasets. Broader evaluations would better demonstrate robustness and real-world applicability.

7. *Heuristic hyperparameter choices:* The weighting coefficient \lambda is heuristically set. While ablation studies are presented, the sensitivity to key parameters (e.g., \lambda, similarity threshold \gamma, trajectory length T) is not thoroughly examined. Such analysis is essential for understanding stability and reproducibility.

8. *Limited comparison to recent baselines:* The study omits relevant preference-based or multi-turn RL frameworks such as RAGEN (arXiv 2025) and LIPO (Liu et al., 2024b).

9. *Weak generalization evidence:* Although the paper claims cross-task generalizability, it provides limited empirical support for transfer to unseen molecular domains. Demonstrating such adaptability would strengthen the claim that POLO constitutes a broadly generalizable paradigm.

[1] Shani, Lior, et al. "Multi-turn reinforcement learning with preference human feedback." NeurIPS 2024

[2] "Multi-turn Training with basic human feedback helps little on LLM reasoning",  arXiv 2025

**Questions:**

See weakness

---

> ### Author Response · Authors · 2025-11-23
> **Response to Reviewer ZM6i Part 1/2**
>
> We thank the reviewer for their feedback. We notice there might be some misunderstandings regarding the content of our paper. We clarify these points below with specific references.
>
>  ## W1: Baseline Comparisons
> The reviewer expressed concern regarding the lack of Bayesian or GPT-based baselines. We respectfully point out that **these comparisons are included in our evaluation**, as detailed in **Section 3.1** and **Table 1**:
>
> *   **Bayesian Optimization:** We compare against **QMO (Query-based Molecule Optimization)**, a representative framework for optimization in latent space.
> *   **GPT-based and LLM-based Methods:** We compared against a wide array of LLM approaches. This includes MOLLEO, which utilizes the BioT5 model (GPT-based Model), as well as six other LLM-based systems ranging from general-purpose models (Qwen, Llama) to chemistry-specific ones (GeLLM3O, ChemLLM).
>
> This comprehensive set of baselines covers both traditional and modern LLM-based approaches in our field. Our SOTA claim is based on POLO's superior performance against these methods under an identical budget of 500 oracle calls.
>
> Regarding the suggestion to include ''Random Forest'' as a baseline: We hope to clarify that Random Forest is a **discriminative model** (used for property prediction), not a **generative model**. It cannot propose new molecular structures or modifications. Therefore, it cannot serve as a baseline for the generative lead optimization task.
>
>
>  ## W2: Novelty Relative to Shani et al.
> We appreciate the reviewer for pointing out this paper. In fact, we have cited this work in our paper. We would like to clarify the fundamental differences that distinguish our contribution from this prior work:
>
> *   **Algorithmic Distinction:** The core difference lies in how preference signals are constructed and utilized.
>     *   Shani et al. rely on **sparse, inter-trajectory preferences**, comparing entire dialogue trajectories (''Dialogue A,  Dialogue B'').
>     *   Our key innovation is extracting **dense, intra-trajectory preferences**. POLO's dual-level learning mechanism compares intermediate steps **within** a single trajectory. This allows us to extract training signals from every intermediate modification, maximizing sample efficiency.
>
> *   **Problem Setting Distinction:** Shani et al. address dialogue systems with human feedback, while we optimize molecules using oracle evaluation, which is a different problem requiring different solutions.
>
>  ## W3 & W4: Algorithm Design and Theoretical Justification
>
> We appreciate the reviewer's thoughtful comments on our algorithm design. While we understand the concern regarding potential conflicts, our paper already provides theoretical and empirical evidence to support the synergy of these components, which we further reinforce with new analysis below.
>
> **1. Theoretical Analysis:** **Appendix I** provides formal analysis of our signal amplification mechanism of PGPO, which shows that our method can extract O($NT^2$) preference signals compared to standard O($N$), thereby maximizing learning from each expensive oracle call.
>
> **2. Empirical Evidence of Synergy:**
> *   **Performance-Level  (Existing):** Our ablation study (**Table 3**) shows that removing either component degrades performance by 10-25%, confirming that the two objectives are complementary, not conflicting.
> *   **Gradient-Level (New Analysis):** To further address the concern about ''conflicting optimization directions,'' we conducted a gradient alignment analysis. This revealed a **cosine similarity of 0.954** between the PPO and preference gradient vectors, demonstrating strong alignment rather than conflict.
>
> **3. Liu et al. Support Our Approach:**
> We thank the reviewer for pointing to Liu et al. In fact, their findings **support the necessity of our multi-turn framework**. That paper argues that multi-turn training provides limited gains for ''complete-information'' tasks with single correct answers. However, molecular optimization is fundamentally different: it is an **incomplete-information exploration problem** where the optimal solution is unknown. Therefore, the limitations observed by Liu et al. do not apply here. Instead, the iterative nature of our problem necessitates a multi-turn approach to progressively refine molecules.

---

> ### Author Response · Authors · 2025-11-23
> **Response to Reviewer ZM6i Part 2/2**
>
> ## W5: On Related Work Coverage
>
> We would like to point out that relevant studies are already covered in our paper. Our **Section 4** provides comprehensive coverage of molecular optimization methods, with **Section 4.2** specifically dedicated to LLM-based approaches.
>
> Regarding ''GPT-based'' methods: We believe there may be a terminology misunderstanding. In our evaluation, our work extensively compares with transformer-based and LLM-based approaches:
> *   MOLLEO (using GPT-based models like BioT5)
> *   Multiple LLM-based methods, including GeLLM³O, ChemLLM, LlaSMol, PEIT-LLM, and others
>
> Since our framework itself is LLM-based, we intentionally prioritize comprehensive coverage of state-of-the-art LLM approaches for molecular optimization, which we view as the most relevant comparison category for our contributions.
>
> That said, we are happy to incorporate any additional references the reviewer recommends to further strengthen the paper’s positioning.
>
>
>
>
>  ## W6: On Evaluation Diversity
>
> We respectfully clarify that our evaluation is comprehensive and follows established standards for this domain.
>
> **1. Standard Benchmarks:**
> ZINC-250k is the canonical dataset for molecular optimization, used in major benchmarks (e.g., PMO, NeurIPS 2022) and by all our baselines (e.g., Reinvent, MOLLEO). Adhering to this standard ensures rigorous, fair head-to-head comparison.
>
> **2. Task Diversity:**
> We evaluate on **10 distinct tasks** (5 single-property, 5 multi-property) spanning physicochemical properties (QED, plogP) and biological activities (DRD2, JNK3), covering a wide spectrum of optimization complexity.
>
> **3. Nature of the Problem:**
> Regarding the request for ''larger datasets'': We emphasize that **Lead Optimization is an instance-based local search problem**, where the goal is to refine specific lead compounds. It is not a dataset-level generation task. Our evaluation across **200 diverse starting leads** provides a statistically robust assessment of the method's ability to handle various chemical starting points.
>
>
>
>  ## W7: Hyperparameter Analysis
>
> We respectfully point out that the specific analyses requested by the reviewer are **already presented in our paper**.
>
> **Section 3.4 (Figure 4)** dedicates significant space to a systematic investigation of trajectory length ($T$) and similarity threshold ($\gamma$). Furthermore, the sensitivity of the preference weight ($\lambda$) is thoroughly analyzed in **Appendix L (Figure 6)**. These analyses span multiple pages and demonstrate that our parameter choices are empirically validated and robust, rather than ''heuristic.''
>
>
>
>  ## W8: On Comparison to Recent Baselines
>
> We thank the reviewer for the suggestions. In fact, both `RAGEN` and `LIPO` are already cited in our paper.
>
> However, these were not included as experimental baselines because they are **general-purpose RL/alignment algorithms**, not specialized methods for molecular optimization. Direct comparison would require significant domain adaptation, making them less meaningful baselines than established domain-specific methods. Our evaluation rightly focuses on methods optimized for the molecular domain to ensure a fair and rigorous assessment of the state-of-the-art.
>
> We will further expand the discussion of these works in the Related Work section to clarify their relationship to our framework.
>
> ## W9: Evidence for Generalization
>
> We would like to highlight the strong generalization evidence already presented in **Appendix K**, which includes two comprehensive studies:
>
> 1.  **Prompt Generalization (Table 9):** Our model maintains **97% performance** on out-of-distribution prompts, demonstrating that it learns the underlying optimization task rather than overfitting to specific phrasing.
> 2.  **Cross-Task Generalization (Table 10):** The model exhibits meaningful knowledge transfer to unseen optimization tasks (e.g., achieving 76% success rate when transferring from JNK3 to QED), proving it acquires generalizable chemical strategies.
>
> These results provide robust empirical evidence that POLO constitutes a generalizable paradigm.

---

> ### Comment · Reviewer_ZM6i · 2025-11-28
>
> I thank the authors for the responses, which addressed some of my earlier concerns. I have adjusted certain points accordingly. However, my major concerns still remain:
>
> ## 1. Baselines and Related Work Coverage
>
> You claim state-of-the-art performance, yet several baselines used in the paper are not representative of the current state of the art—especially the non-LLM baselines. Outperforming these older or weaker baselines is not sufficient to support an SOTA claim.
>
> ### Counterfactual 1: Bayesian Optimization
> You write:
>
> > “Bayesian Optimization: We compare against QMO (Query-based Molecule Optimization), a representative framework…”
>
> However:
> * **QMO is not a Bayesian optimization approach.** It is a zeroth-order (gradient-free) optimization method.
> * The QMO paper does not describe itself as Bayesian optimization.
> * QMO is from 2019, and many more recent and stronger optimization baselines exist today.
>
> ### Counterfactual 2: REINVENT Baseline
> In your paper (Page 19, line 1003):
>
> > “REINVENT 4 (Loeffler et al., 2024) is a reinforcement learning approach using an **RNN-based** agent.”
>
> This is factually incorrect:
> * **REINVENT 2 (2019)** uses an RNN for SMILES generation.
> * **REINVENT 4 (2024)** uses a Transformer / GPT-style model and significantly outperforms REINVENT 2.
> * **Even so, REINVENT 4 is not SOTA. Many recent works (cited REINVENT 4)—both published or on arXiv—surpass REINVENT 4 with significance.** None of these closely related works are discussed, cited, or compared against in your paper, which weakens the SOTA claim.
>
> ---
>
> ## 2. Machine Learning Baselines
>
> My earlier comment was not specifically about random forests—it was about the broader absence of core machine-learning–based drug optimization baselines. Random forests were simply an illustrative example. In fact, even random forest can perform reasonably well. For instance:
>
> > Liu, Jiajia, et al. “Application of random forest based on semi-automatic parameter adjustment for optimization of anti-breast cancer drugs.” *Frontiers in Oncology*, 2022.
>
> A more representative SOTA of ML baselines is needed for a rigorous comparison. My concern is that when the optimization objective is clearly defined—for example, optimizing QED—the additional conversational component introduced by an LLM may actually hinder the optimization process by introducing additional complexity.
>
> ---
>
> ## 3. Algorithm Design and Novelty
>
> Regarding the design choice of turn-wise vs. trajectory-wise preference signals:
>
> In **Shani et al.** (page 4), the authors explicitly discuss both settings.
> * They note that their algorithm can be adapted to turn-wise feedback.
> * They intentionally choose trajectory-wise feedback, which **aligns with the concerns raised in my earlier comment.**
>
> Their motivating example is:
>
> > A selling agent asking for a high price may seem immediately bad, but becomes good as part of a complete long-term strategy.
>
> This maps directly to molecular optimization. Your method uses turn-level molecule proposals as feedback. While this increases signal frequency, these intermediate signals may introduce substantial noise instead, potentially harming policy improvement.
>
> In reinforcement learning:
> * Agents often must traverse temporarily low-reward regions to reach globally optimal solutions;
> * Intermediate “suboptimal” molecules may be essential for exploring promising chemical space;
> * Overly penalizing such steps can impair long-horizon optimization.
>
> Thus, turn-wise supervision risks discouraging beneficial exploratory steps and may hinder convergence toward globally optimal molecular designs—a concern reinforced by my prior empirical findings on similar domain.
>
>
> Respectfully point out that these **major concerns** remain and warrant further attention. Thanks.

---

### Meta-Review · Area_Chair_x2fy · 2025-12-26

**Summary:**

This paper mainly focuses on drug discovery tasks in the context of large language models. The paper points out that existing works in this domain suffer from sampling inefficiency. Thus, this paper proposes to leverage the in-context learning and instruction following natures of large language models to solve such an issue. To this end, an optimization method, preference-guided multi-turn optimization for lead optimization (POLO), is proposed to take advantage of both turn-level and trajectory-level information for optimization to enhance the performance.
According to the reviews of this paper, all reviewers consistently decided to reject this paper since the paper has several drawbacks:
- **The drawback of the framework.** According to Reviewer xJaw, the operation that expresses the comparisons in a numerical way may lead to the loss of information. Meanwhile, the Reviewer LZd8 pointed out that the proposed method relies on the synthetic data, which implies that LLM may mimic the behaviours in these data.
- **The Unfair Comparisons.** According to Reviewer LZd8, the success rate on JNK3 is 0 in this paper, while the results in the original MOLLEO paper are good. Meanwhile, compared to the proposed method, which is trained on task-specific tasks, the previous methods are not trained on specific tasks. This raised the concern of unfair comparison. Moreover, Reviewer xJaw mentioned that the oracle budgets are fewer than the PMO benchmark, which may negatively affect the performance.
- **The Incomplete Evaluation.** According to Reviewer xJaw, the tasks adopted in this paper are relatively simple. Meanwhile, since LLMs might be exposed to these oracles, the evaluation of this paper may be incomplete.
In general, the paper is well written and organized. The quality of this submission is quite good. However, due to the concerns unsolved above, we encourage the authors to further polish their paper and resubmit it to other venues.

**Reviewer Concerns:**

I have mentioned the concerns that remain unsolved in the summary part above. For other concerns regarding this paper, such as double counting and credit assignment, I think they have been addressed through rebuttal.

**Reviewer Scores:**

From the rebuttal records of this submission, I think that Reviewer LZd8 would increase the score, while other reviewers may maintain their scores.

---

### Decision · Program_Chairs · 2026-01-26

Reject